# A neurocomputational account of the link between social perception and social action

Lisa M Bas[1]*[†], Ian D Roberts[2][†], Cendri A Hutcherson[2,3][‡], Anita Tusche[1,4]*[‡]

[1]Department of Psychology, Queen's University, Kingston, Canada; [2]Department of Psychology, University of Toronto Scarborough, Toronto, Canada; [3]Department of Marketing, Rotman School of Management, University of Toronto, Toronto, Canada; [4]Center for Neuroscience Studies, Queen's University, Kingston, Canada

### eLife assessment

These **important** findings stand out from other similar studies via some **convincing** demonstration of behavioural and neural relationships between two helping tasks – one focusing more on social perception, one more on its influence on social behaviour – that were performed more than 300 days apart. The claims however would be stronger with a larger sample size.

*For correspondence:
lisa.bas@queensu.ca (LMB);
anita.tusche@queensu.ca (AT)

[†]These authors contributed equally to this work
[‡]These authors also contributed equally to this work

Competing interest: The authors declare that no competing interests exist.

**Abstract** People selectively help others based on perceptions of their merit or need. Here, we develop a neurocomputational account of how these social perceptions translate into social choice. Using a novel fMRI social perception task, we show that both merit and need perceptions recruited the brain's social inference network. A behavioral computational model identified two non-exclusive mechanisms underlying variance in social perceptions: a consistent tendency to perceive others as meritorious/needy (bias) and a propensity to sample and integrate normative evidence distinguishing high from low merit/need in other people (sensitivity). Variance in people's merit (but not need) bias and sensitivity independently predicted distinct aspects of altruism in a social choice task completed months later. An individual's merit *bias* predicted *context-independent* variance in people's overall other-regard during altruistic choice, biasing people toward prosocial actions. An individual's merit *sensitivity* predicted *context-sensitive* discrimination in generosity toward high and low merit recipients by influencing other- and self-regard during altruistic decision-making. This context-sensitive perception–action link was associated with activation in the right temporoparietal junction. Together, these findings point toward stable, biologically based individual differences in perceptual processes related to abstract social concepts like merit, and suggest that these differences may have important behavioral implications for an individual's tendency toward favoritism or discrimination in social settings.

## Introduction

Psychologists and economists have long sought to explain when and why people help. While people are generally altruistic, they also show selectivity, being more likely to assist those in need (***Batson et al., 1995***; ***Batson et al., 2007***; ***Cappelen et al., 2013***; ***Molouki and Bartels, 2020***; ***Batson et al., 1986***) and to withhold aid from those perceived as undeserving (***Hare et al., 2010***; ***Tusche et al., 2016***; ***Fong, 2007***; ***Eckel and Grossman, 1996***; ***Van Doesum et al., 2022***). Need and deservingness (merit) are two distinct principles of morality. The need principle involves distributing resources to those requiring them, regardless of whether they have earned them. In contrast, the 'merit principle'

**eLife digest** The question of when and why humans are more likely to lend each other a hand has long been studied. In general, we are more likely to help those who appear more needy or deserving of help. This explains why there is often an outpouring of support after natural disasters and why people are more likely to assist children, who are perceived as being more 'innocent.'

Unfortunately, people's spontaneous judgments of need and merit can lead to adverse outcomes. For example, racial stereotyping influences our perceptions of who deserves support. Additionally, different people assess deservingness differently, leading to individual differences in whom we help.

Bas, Roberts et al. wanted to understand how people assess whether others need or deserve help and how this influences their decision to offer assistance (or not). To answer these questions, the researchers examined two competing hypotheses: some individuals may be more or less inclined to perceive others as needing and deserving help, regardless of the context. Alternatively, some people may be more sensitive to contextual cues signaling merit or need. These two processes could explain individual differences in how humans perceive need and deservingness, influencing decisions to help.

Bas, Roberts et al. developed a model of social perception that encompassed both of their hypotheses and designed a laboratory task that tested how social perceptions influence helping behavior. This approach measured two potential mechanisms for how social perception might influence choices: individuals could have a general bias that determines how inclined they are to perceive others as deserving or needy, or differences in sensitivity to integrating context-specific cues. Intriguingly, both mechanisms predicted their willingness to help strangers months later in a separate altruism task.

People who were biased towards perceiving others as deserving paid more attention to others' welfare when they had a chance to help and were more altruistic. People who were more sensitive to context cues around deservingness were more likely to discriminate between others, assisting those perceived as deserving help while withholding aid from supposedly non-deserving individuals. This aspect of the perception-action link was related to brain activation in the right temporoparietal junction, an area of the brain crucial to making judgments about others.

Bas, Roberts et al.'s findings point towards biological differences in how people perceive abstract social concepts like merit or deservingness. The way people perceive these concepts is stable and influences altruistic choice behavior. These results suggest that altruism may be increased by changing how people perceive others, leading to reduced favoritism or discrimination in social settings.

focuses on allocating resources based on individuals' deservingness, irrespective of their actual need (*Wilson, 2003*). How do people assess whether others deserve or need help, and how does this influence their choices to initiate that help?

Based on an analogy to perception in basic sensory domains like vision (*Green and Swets, 1966*; *LaBerge et al., 1969*), we hypothesized that variance in social perceptions might be driven by two discrete mechanisms: an individual's *sensitivity* to social situational cues signaling other's merit or need, and an individual's *bias* to perceive merit or need independently of specific cues. In other words, one can think of the likelihood $P$ that person $J$ perceives a target individual as deserving (meritorious) or in need of aid as being determined by the sum of all cues $C_1, C_2, \ldots C_N$ associated with that perceptual judgment, multiplied by a person's idiosyncratic sensitivity $S$ to each cue, and added to their baseline tendency (*bias*) to perceive others as deserving or in need:

$$P(Judgment|J, C) \sim \sum_{i=1}^{N} S_{i,J}C_i + bias_J \qquad (1)$$

Need-signaling cues could include facial, vocal, or postural cues of pain or distress (*FeldmanHall et al., 2015*; *Murray, 1979*; *Marsh, 2016*; *Small and Verrochi, 2009*), or cues implying imminent harm (e.g., a person in front of a runaway car; *Rand and Epstein, 2014*; *Vieira et al., 2020*). Merit-relevant cues could include membership in 'good' or 'bad' groups like children (*Burt and Strongman, 2005*), one's in-group (*Stürmer et al., 2005*), Nazi Party membership (*Opotow, 1990*), or information about the social normativity or benevolence of that person's actions (*Van de Vondervoort et al., 2018*). An individual with higher average *sensitivity* to the cues associated with that judgment would

discriminate more between those who are high or low in merit or need (as signaled by available cues). An individual with a larger *bias* term would simply be more likely to judge *all* individuals as meritorious/needy irrespective of present cues. Whether processing and integrating these social cues, along with the resulting judgments, involves distinct neural modules (*Santavirta et al., 2023*) or a general-purpose mechanism for social cognition (*Thornton and Mitchell, 2018*), such as the mentalizing network (*Van Overwalle, 2009*; *Schurz et al., 2014*; *Molenberghs et al., 2016*; *Bzdok et al., 2012*), remains unknown.

A similar question arises over *how* perceptions of need and merit influence meaningful *behavior*. While extensive research shows that they do (*Cappelen et al., 2013*; *Molouki and Bartels, 2020*; *Batson et al., 1986*; *Hare et al., 2010*; *Fong, 2007*; *Eckel and Grossman, 1996*; *Van Doesum et al., 2022*; *Engel, 2011*; *Jilke and Tummers, 2018*; *Batson et al., 1996*), we know little about the precise computational mechanism underlying this perception–action link. We speculated that it might operate by changing the value that people ascribe to different social considerations during choice. More specifically, we build on a growing body of work suggesting that prosocial actions can be characterized as a value-based decision process (*Tusche and Bas, 2021*; *Chen et al., 2024*; *Teoh et al., 2020*; *Mayr and Freund, 2020*). In this framework, the value of acting prosocial is represented as a weighted sum of several attributes, like the cost of helping for the decision-maker, the benefits for others, or the fairness of the outcome. These three attributes have been repeatedly shown to guide social behaviors (*Morishima et al., 2012*; *Fehr and Schmidt, 1999*; *Charness and Rabin, 2002*; *Tricomi et al., 2010*; *Tabibnia et al., 2008*; *Rhoads et al., 2021*; *Carlson et al., 2020*), with higher weights on prosocial attributes like benefits for others or fairness yielding more prosocial choices (*Teoh et al., 2020*; *Saulin et al., 2022*; *Hutcherson et al., 2015a*; *Hu et al., 2023*; *Hu et al., 2021*; *Sul et al., 2015*; *Li et al., 2022*). Within this computational framework, perceptions of others' merit or need could affect social behaviors through a simple mechanism: by altering the weights given to outcomes for self, outcomes for others, or fairness during the decision process. For example, perceiving someone as highly deserving or needy could increase helping by increasing the weight given to the recipient's benefits, decreasing the weight given to one's own benefits, or both.

This computational framework leads to a set of testable predictions about how individual differences in social perceptual sensitivity and bias (*Equation 1*) might influence altruistic behavior: An individual's general tendency to perceive merit or need (i.e., their *bias* parameter) should be correlated with a general tendency to ascribe higher weight to prosocial considerations (i.e., others' benefits or fairness) across settings. In contrast, an individual's sensitivity to cues signaling merit or need (i.e., their average *sensitivity* parameter *S*) should make them more discriminatory, increasing weights on prosocial considerations for those judged as deserving (or needy) but decreasing them for those judged as lacking these qualities. These weights should then manifest in the frequency with which an individual acts generously, either overall or as a function of the recipient's merit and/or need.

To test these hypotheses and uncover their neural basis, we first asked participants to complete a novel social perception task while we collected their brain responses using fMRI (functional magnetic resonance imaging). The behavior observed in this task allowed us to computationally estimate two distinct—but not mutually exclusive—processes underlying individual differences in merit/need perception: an overall propensity to perceive others as deserving/in need (i.e., an individual's *bias* parameter) and a tendency to sample and integrate cues signaling merit/need (i.e., an individual's *sensitivity* parameter). Neurally, we investigated whether merit and need are perceived through distinct or common neural circuits and reflect individual differences in people's perceptual bias and/or sensitivity. Examining need and merit concurrently in this task also helped clarify the computational and neural underpinnings of these related but distinct concepts, distinguishing between them more effectively. Second, we used a separate altruistic choice task (completed on average ~303 days later) to computationally estimate the weights people place on themselves, others, and fairness when deciding whether to provide aid, and we examined how those weights varied as a function of recipient merit and need. Third, we tested whether individual differences in merit/need bias or sensitivity and their neural underpinnings in the social perception task predict individual differences in prosocial action in the altruism task. Our results suggest that merit (but perhaps not need) perceptions result from stable individual differences in both bias and sensitivity and manifest in an individual's prosocial behavior months later.

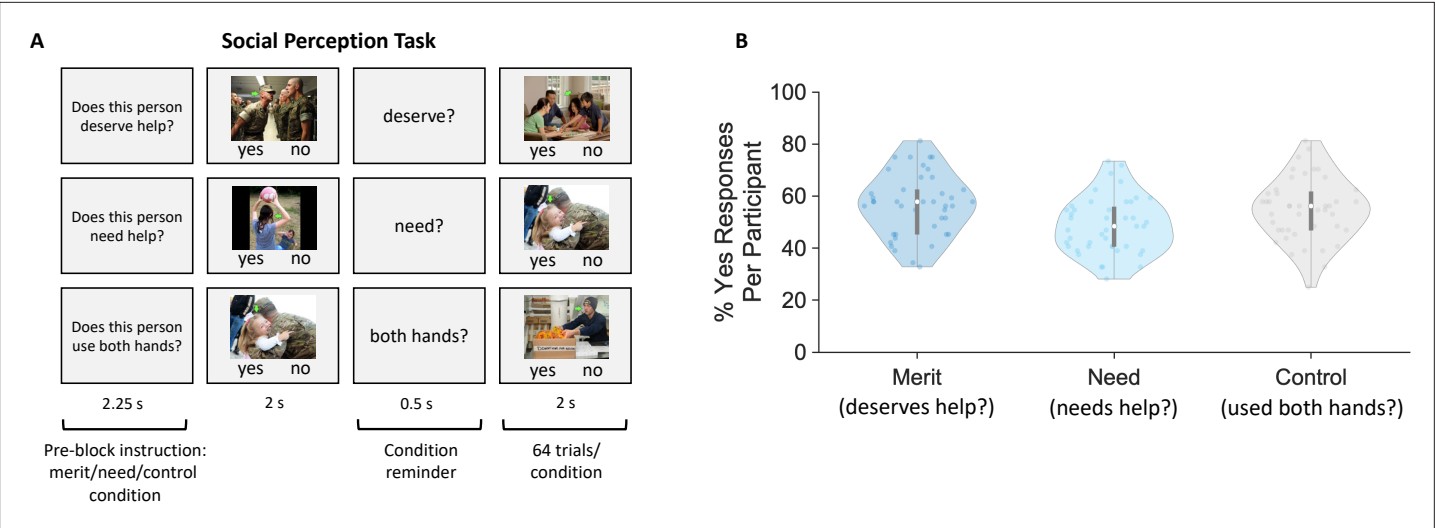

**Figure 1.** Social perception task and perceptual judgments. (**A**) Social perception task. Participants made rapid yes/no judgments regarding others' perceived deservingness (merit blocks, top row), need (need blocks, middle row), or factual inferences (control blocks, bottom row) while their brain responses were measured using fMRI. All photographs included in this panel are in the public domain in the United States. (**B**) Social perception responses. Violin plots of the participant-specific percentage of yes responses (dots) for merit (left), need (middle), and control judgments (right). Merit: 56.70 ± 11.91; Need: 48.69 ± 10.81; Control: 54.87 ± 12.03. Edges of boxplots (gray bars) indicate the 25th–75th percentiles, boxplot whiskers illustrate minima and maxima, and central white dots represent median values. Two outliers for merit and one outlier for need responses were excluded based on values that exceeded 3 standard deviations from the mean.

## Results

This section is structured as follows. First, we report the results of our novel fMRI social perception task to describe how people perceive others' merit and need. Behaviorally, we used parameter estimates from our computational model of social perception to characterize individuals' perceptual biases and sensitivities driving variance in social perceptions across people. Neurally, we localized brain regions recruited during merit and need inferences and asked whether neural computations underlying merit and need perceptions are supported by distinct neural circuits or a general-purpose network for social inference processing. Second, we used data from a separate altruism task (collected on average 303 days later) to describe the effects of an interaction partner's perceived merit and need on meaningful social behavior. Third, we explored if individuals' perceptual biases and sensitivities (as captured in the social perception task) predicted variance in actual social behavior (observed in the altruism task). We examined this core question about the social perception–action link across time and contexts at both the behavioral and neural levels.

### Social perception task: behavior and neural underpinnings of need and merit inferences

To assess individual differences in social perceptions, we used a modified version of the established fMRI why/how task (*Spunt and Adolphs, 2014*; *Spunt and Adolphs, 2015*). Participants viewed images of people in real-world scenes and made rapid yes/no judgments about perceived merit, need, or factual attributes (control condition) in separate blocks while their brain responses were measured (*Figure 1A*). Despite viewing the same stimuli, participants differed dramatically in their judgments of others' merit and need (*Figure 1B*). The percentage of trials perceived by participants as depicting someone as deserving (merit blocks) ranged from 33% to 81%. Similarly, the range for perceiving someone in need (need blocks) was 28–73%.

### Computational behavioral model of social perception

What mechanisms drive these profound individual differences in social perceptions (*Figure 1B*)? To address this question, we examined two distinct computational mechanisms: first, people may differ in their general tendency to perceive others as deserving or in need (bias hypothesis). Second, people

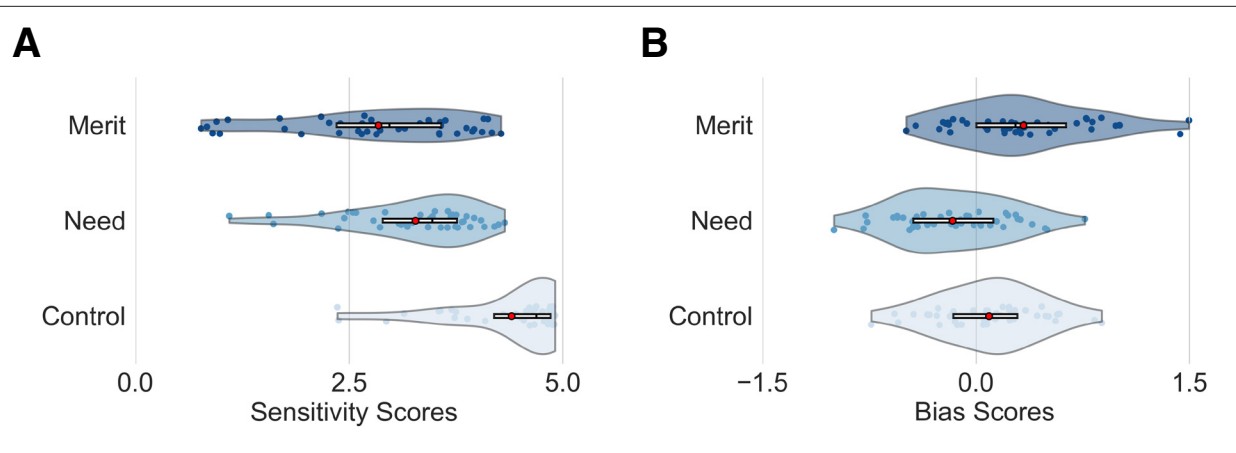

**Figure 2.** Estimates of the computational model of social perception. (**A**) Violin plots illustrate the distribution of participant-specific estimates (dots) of the perceptual sensitivity and (**B**) perceptual bias for each inference condition in the social perception task (merit, need, and control). Edges of boxplots (bars) indicate the 25th–75th percentiles; central red dots represent mean values.

The online version of this article includes the following figure supplement(s) for figure 2:

**Figure supplement 1.** Model fit of our computational model of social perception illustrated via the correspondence between predicted (model) and observed (data) choices (left panel) and reaction times (RTs; middle and right panel).

**Figure supplement 2.** Intercorrelation of estimates of the computational behavioral model of social perception (upper values indicate Spearman's correlation coefficients, lower values represent p-values, FDR corrected for multiple comparisons, and pairwise outlier exclusions based on ± 3 standard deviations; 1.52% outliers removed for all variables).

may differ in their perceptual sensitivity to merit- or need-related evidence in their choice environment and/or their ability to use this information to guide their judgments (sensitivity hypothesis). Importantly, these mechanisms are not mutually exclusive but may vary in their relative impact on social behaviors across people and contexts. We tested the contribution of these two potential mechanisms using our behavioral computational model of social perception (see Methods).

We first verified that the model adequately accounted for the differences in participants' choices and response times in all three task conditions (merit, need, and control blocks; for an illustration of the model fit, see *Figure 2—figure supplement 1*).

Next, we examined the estimated parameters of our hierarchical computational model at the individual participant level. Positive average sensitivity parameters for merit ($S_{merit}$ in merit blocks = 2.84 ± 1.02, mean ± std, significantly different from zero, p < 0.001, FDR corrected), need ($S_{need}$ in need blocks = 3.28 ± 0.75, p < 0.001, FDR corrected), and control ($S_{control}$ in control blocks = 4.40 ± 0.69, p < 0.001, FDR corrected) verified that participants accurately distinguished between targets who deserve help, need help, and use both hands, and used this perceptual evidence to guide their judgments (*Figure 2A*). For the estimated perceptual bias parameters, people tended to perceive others as deserving (indicated by an average positive merit bias, $Bias_{merit}$ = 0.33 ± 0.47, p < 0.001, FDR corrected) and not in need of help (indicated by an average negative need bias, $Bias_{need}$ = –0.17 ± 0.39, p = 0.009, FDR corrected). In the control condition, the estimated bias showed a small positive but non-significant trend ($Bias_{control}$ = 0.09 ± 0.36, p = 0.099, FDR corrected; *Figure 2B*). It is worth noting that the *Bias* parameters are strongly associated with (though not the sole determinant of) the mean response rate. For a description of the estimated values for the hyper-mean parameters in our model, see *Supplementary file 1*.

Notably, individuals' bias and sensitivity estimates were uncorrelated for merit (Spearman's r = 0.10, p = 0.73, FDR corrected; $S_{merit}$ and $Bias_{merit}$) and need judgments (Spearman's r = –0.10, p = 0.73, FDR corrected; $S_{need}$ and $Bias_{need}$), suggesting two distinct mechanisms driving variance in social perceptions across people. Likewise, we found no link between need and merit sensitivity (Spearman's r = –0.22, p = 0.60, FDR corrected; $S_{merit}$ and $S_{need}$), indicating that people might be sensitive to cues signaling merit but not need, and vice versa. However, estimates of the need bias and merit bias were correlated (Spearman's r = 0.73, p < 0.001, FDR corrected; $Bias_{merit}$ and $Bias_{need}$), suggesting that people who tend to perceive others as needy might also tend to perceive others as deserving.

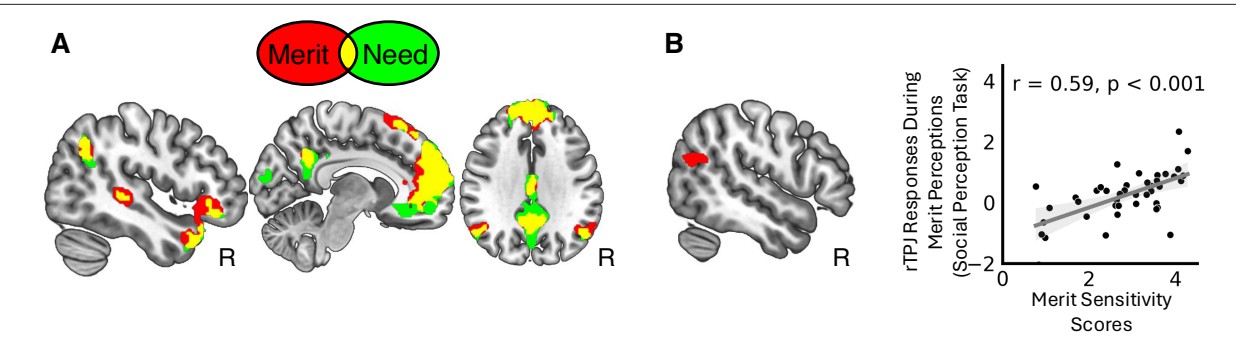

**Figure 3.** Neural correlates of need and merit inferences. (**A**) Need and merit inferences activate the mentalizing network to a similar extent (red illustrates brain regions activated for [merit − control], green illustrates brain regions identified for [need − control], and yellow indicates overlap; both contrast maps thresholded at p < 0.001 at the voxel level, family-wise error (FWE) corrected at the cluster level at p < 0.05; R = right hemisphere). (**B**) Activity in the right temporoparietal junction (TPJ) during merit perceptions [merit − control] reflects individual differences in merit sensitivity scores estimated in the computational model of social perception.

The online version of this article includes the following figure supplement(s) for figure 3:

**Figure supplement 1.** Conjunction of brain areas recruited during need and merit perceptions (social perception task) of brain maps identified for [need − control] and [merit − control] inferences (each social perception map thresholded at p < 0.001 at the voxel level, family-wise error (FWE) corrected at the cluster level at p < 0.05).

**Figure supplement 2.** Cluster in the right temporoparietal junction (rTPJ) activated during merit perceptions (social perception task) that reflect individual differences in merit sensitivity estimated in the computational model of social perception (in only the n = 25 participants with overlapping altruistic choice task data).

See *Figure 2—figure supplement 2* for full details on intercorrelations of estimates of the behavioral computational model. These findings highlight the benefits of formal computational models, which can capture and quantify distinct processes that can be hard to distinguish based on observed behaviors alone. It also raises the interesting question of whether the specificity of need and merit observed at the behavioral level gives rise to inference specificity at the neural level.

## Neural underpinnings of merit and need perceptions

Next, we examined the neural substrates of need and merit inferences obtained in the social perception task. The task is a modified version of an established fMRI why/how task (*Spunt and Adolphs, 2014*; *Spunt and Adolphs, 2015*) that reliably identifies the mentalizing network (*Straccia et al., 2023*; *Tan et al., 2019*; *Thompson et al., 2022*; *Cardenas et al., 2021*; *Tusche et al., 2023*). Not surprisingly, we found that need and merit inferences also recruited the mentalizing network. The medial prefrontal cortex, superior temporal sulcus, temporal pole, temporoparietal junction (TPJ), and posterior cingulate cortex were reliably activated during both merit and need inferences, among other regions. *Figure 3A* illustrates the overlap of brain areas activated during merit and need judgments. *Table 1* provides the results of the condition-specific brain maps and the formal conjunction of brain areas identified for [need − control] and [merit − control] inferences, each thresholded at p < 0.001 at the voxel level, family-wise error (FWE) corrected at the cluster level at p < 0.05 (see *Figure 3—figure supplement 1* for a visualization of the conjunction analysis). No brain region was significantly more activated during merit than need inferences [merit − need] at our omnibus threshold. However, for the reverse contrast [need − merit], need inferences showed significantly greater activation in the cuneus (MNI: [−4, −98, 18], 8741 voxels, t = 7.64), intraparietal sulcus (MNI: [28, −62, 56], 256 voxels, t = 4.58), and sensorimotor cortex (MNI: [4, −38, 64], 294 voxels, t = 4.39) (p < 0.001 at the voxel level, FWE corrected at the cluster level at p < 0.05), suggesting an enhanced activation of the (extended) mirror system (*Pineda, 2008*; *Van Overwalle and Baetens, 2009*). Future research may reveal additional distinctions between merit and need appraisals in trial-wise (compared to our block-wise) fMRI designs.

We also employed supplemental multivariate decoding analyses (searchlight analysis; *Kriegeskorte et al., 2006*; *Haynes and Rees, 2006*; *Heinzle et al., 2012*), as commonly used in social perception and neuroscience research (*Tusche et al., 2016*; *Tusche et al., 2023*; *Brooks et al., 2021*; *Tusche*

**Table 1.** Neural activations during social need and merit inferences in the social perception task.

| Brain region | Side | T | k | MNI (peak) | | |
|---|---|---|---|---|---|---|
| | | | | x | y | z |
| *[Need − control] ∩ [Merit − control]* | | | | | | |
| Dorsomedial prefrontal cortex | L | 7.78 | 6250 | −14 | 38 | 48 |
| Superior temporal sulcus | L | 6.61 | 1154 | −64 | −26 | −14 |
| Superior temporal sulcus | R | 5.42 | 146 | 70 | −32 | 0 |
| Temporal pole | R | 6.16 | 523 | 40 | 22 | −32 |
| Posterior cingulate cortex | L | 5.97 | 671 | −2 | −54 | 32 |
| Midcingulate cortex | L | 5.17 | 119 | −2 | −10 | 34 |
| Temporoparietal junction (TPJ) | R | 4.56 | 167 | 54 | −66 | 34 |
| TPJ/angular gyrus | L | 4.58 | 141 | −50 | −62 | 32 |
| Cerebellum | L | 5.33 | 336 | −26 | −82 | −42 |
| *[Merit > control]* | | | | | | |
| Medial prefrontal cortex | L | 8.04 | 7616 | −8 | 54 | 18 |
| TPJ/ angular gyrus | R | 4.96 | 278 | 46 | −58 | 34 |
| TPJ/angular gyrus | L | 4.54 | 267 | −60 | −60 | 34 |
| Superior temporal gyrus | R | 5.84 | 526 | 68 | −34 | −2 |
| Posterior cingulate cortex | L | 6.56 | 687 | −4 | −48 | 28 |
| Medial cingulate cortex | L | 5.29 | 127 | −2 | −14 | 34 |
| Caudate nucleus | L | 5.43 | 74 | −12 | 10 | 12 |
| Anterior insula | L | 7.61 | 1973 | −38 | 20 | −14 |
| Anterior insula | R | 7.35 | 1123 | 34 | 22 | −16 |
| Dorsolateral prefrontal cortex | L | 4.45 | 66 | −34 | 22 | 46 |
| Cerebellum | R | 4.69 | 95 | 26 | −88 | −34 |
| Cerebellum | L | 6.05 | 471 | −26 | −80 | −34 |
| *[Need > control]* | | | | | | |
| Medial prefrontal cortex | L | 7.78 | 7522 | −14 | 38 | 48 |
| TPJ/angular gyrus | L | 4.58 | 261 | −50 | −62 | 32 |
| TPJ/angular gyrus | R | 4.56 | 252 | 54 | −66 | 34 |
| Medial temporal cortex | L | 6.61 | 1385 | −64 | −26 | −14 |
| Temporal pole | R | 6.16 | 801 | 40 | 22 | −32 |
| Superior temporal cortex | R | 5.42 | 181 | 70 | −32 | 0 |
| Posterior cingulate cortex | L | 5.97 | 1727 | −2 | −54 | 32 |
| Dorsolateral prefrontal cortex | L | 4.86 | 132 | −40 | 22 | 48 |
| Cerebellum | L | 5.33 | 362 | −26 | −82 | −42 |
| Cerebellum | R | 4.77 | 219 | 26 | −86 | −34 |
| Cuneus | L | 5.34 | 284 | −4 | −100 | 18 |
| *[Need − control] > [Merit − control]* | | | | | | |
| Visual cortex | L | 8741 | 7.64 | −4 | −98 | 18 |
| Intraparietal sulcus | R | 256 | 5.24 | 28 | −62 | 56 |

*Table 1 continued on next page*

*Table 1 continued*

| Brain region | Side | T | k | MNI (peak) | | |
|---|---|---|---|---|---|---|
| Somatosensory cortex | R | 294 | 4.97 | 4 | –38 | 64 |

Note. Results are reported at a statistical threshold of p < 0.001 at the voxel level, family-wise error (FWE) corrected at cluster level at p < 0.05. There were no significant results for [Merit – control] > [Need – control]. Only peak activations of each cluster are shown. L = left hemisphere, R = right hemisphere, MNI = Montreal Neurological Institute, *k* = cluster size in voxels.

*and Hutcherson, 2018*; *Corradi-Dell'Acqua et al., 2016*), corroborating our univariate findings (see *Supplementary file 2*). Specifically, these additional analyses confirm the involvement of the mentalizing network in the processing of merit (vs. control) and need (vs. control) inferences. Moreover, these supplemental analyses failed to identify multivariate activation patterns that reliably decoded need versus merit inferences in the social perception task (p < 0.001 at the voxel level, FWE corrected at the cluster level at p < 0.05). In other words, multivoxel activation patterns did not allow decoding of whether participants were currently judging others' need or merit at this statistical level, suggesting common neural codes for both types of social perceptual judgments.

Taken together, our results demonstrate that both need and merit inferences reliably recruited the well-established mentalizing network, and to a comparable extent. Answering our first core question: These findings are consistent with the notion that both appraisals are supported by general-purpose rather than domain-specific social cognitive mechanisms.

## Model estimates of merit sensitivity modulate the neural underpinnings of merit perceptions

So far, we showed that a general-purpose network involved in social inference processes is recruited during both merit and need inferences. Next, we ran a series of whole-brain analyses to identify brain areas for which inference-evoked brain responses covary with individual differences in estimates of the computational model of social perception. Given the evidence above that the social cognitive brain network is activated to make these judgments, we hypothesized that stronger activation in one or more hubs in this network should correlate with greater perceptual sensitivity and/or bias in these social inferences. Consistent with this prediction, we found that brain responses during merit inferences [merit – control] systematically covaried with participants' merit sensitivity scores in the right temporoparietal junction (rTPJ, peak at [MNI 56, –64, 22], *t* = 4.24, *k* = 228 voxels, p < 0.001 at the voxel level, FWE corrected at the cluster level at p < 0.05). In other words, participants with larger merit sensitivity scores (captured in the computational behavioral model) exhibited larger rTPJ responses when appraising someone's merit (*Figure 3B*). This effect was specific for brain responses obtained during merit inferences; an analogous whole-brain analysis using neural activation obtained during need inferences [need – control] to predict merit sensitivity did not yield significant effects. We also did not observe any brain regions where inference-evoked brain responses were associated with variance in participants' need sensitivity or bias scores for either merit or need at the whole-brain level (p < 0.001 at the voxel level, FWE corrected at the cluster level at p < 0.05). Thus, despite merit and need inferences both recruiting the general-purpose mentalizing network to an equivalent extent on average, we found some evidence of neuroanatomical specificity for merit inferences when considering estimates of the computational behavioral model (*Figure 3B*). Notably, this functional link between the rTPJ and merit sensitivity was robust when we repeated the whole-brain analysis for the reduced sample of *n* = 25 participants with overlapping altruism task data (p < 0.001 at the voxel level, FWE corrected at the cluster level at p < 0.05; for details, see *Supplementary file 3*; for illustration, see *Figure 3—figure supplement 2*).

## Altruism task behavior

The results reported above relate to *perceptions* of need and merit; however, they say nothing about how such perceptions might influence decisions to help. In a separate altruism task, we examined how independently manipulating a social target's merit and need alter prosocial behavior. The altruism task required participants to accept or reject monetary allocation that affected their payoffs and that of one of three partners (see *Figure 4A*). The merit of the three partners (high, unknown, and low) was

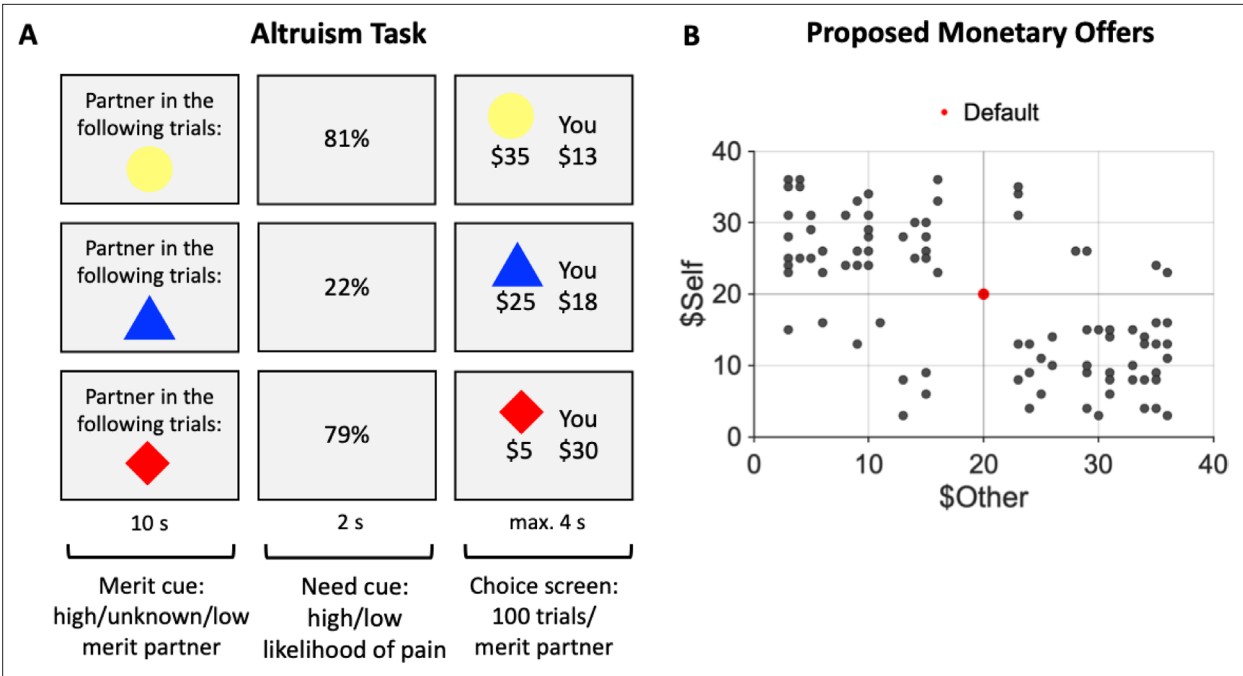

**Figure 4.** Altruism task and its proposed monetary offers. (**A**) Altruism task. On each trial, participants accepted or rejected a monetary offer that affected the payoffs for themselves (You) and one of three partners (choice screen; displayed offer vs. constant default of $20 for both). The three partners (identified via colored geometric shapes) differed in their associated merit (merit cue: high/low/unknown) based on partner behavior in a separate exchange game played before the altruism task (see Appendix 1). Partner's need (need cue: high/low) was manipulated on a trial-by-trial basis, indicated by the likelihood of a painful cold pressor task (CPT) for the partner after the altruism task (high need: 80 ± 4%; low need: 20 ± 4%). Partners could buy out of the painful CPT using funds from one randomly selected trial at the end of the altruism task. Participants were informed that they could help their partners avoid the painful CPT by making generous choices. (**B**) Proposed monetary offers for the participant (You) and the partner (represented by one of three colored geometric shapes in **A**) ranged from $5 to $35. Offers are illustrated as indicated as black dots, whereas the constant default offer ($20 for both) is indicated as a red dot. Monetary proposals were selected such that most choices involved a trade-off between higher payoffs for participants and partners (upper left and lower right quartile), compared to the constant default offer.

The online version of this article includes the following figure supplement(s) for figure 4:

**Figure supplement 1.** Sanity checks of successful merit and need manipulation.

manipulated based on information about partners' behavior in a separate exchange game completed before the altruism task (see Appendix 1). Partners need varied on a trial-by-trial level (high and low) based on the likelihood of performing a painful cold pressor task (CPT) after the experiment. Participants were informed that partners could use the money earned during the altruism task (based on one randomly selected trial) to buy out of the painful post-task CPT. Importantly, the altruism task was unrelated to the social perception task and was completed on average 303 days later. It allowed us

**Table 2.** Generosity in the altruism task (fraction of trials with generous choices).

| Condition | Mean | SD | Min | Max |
|---|---|---|---|---|
| Overall generosity (across conditions) | 0.34 | 0.23 | 0.08 | 0.91 |
| Merit low, Need low | 0.24 | 0.24 | 0.06 | 0.86 |
| Merit low, Need high | 0.30 | 0.30 | 0.00 | 0.98 |
| Merit unknown, Need low | 0.32 | 0.22 | 0.08 | 0.90 |
| Merit unknown, Need high | 0.43 | 0.28 | 0.06 | 0.94 |
| Merit high, Need low | 0.34 | 0.21 | 0.06 | 0.80 |
| Merit high, Need high | 0.44 | 0.29 | 0.08 | 0.96 |

to characterize how people *act* on perceptions of merit and need when deciding whether to give aid to another person.

## Partner's need and merit alter generosity in the altruism task

Did the experimental manipulations of another's need and merit affect people's generosity during altruistic choice? We addressed this question by fitting a mixed-effects logistic regression model to the observed generous and selfish choices (coded as 1/0) in the altruism task (*Equation 3*, see Methods). Choices were classified as generous if the participant *accepted* a proposal that benefited the partner at the expense of oneself ($Self < $Other) or *rejected* a proposal that profited themselves at the partner's expense ($Self > $Other) (see *Table 2* for the summary statistics of the proportion of generous choices and *Figure 5A* for an illustration).

The model's total explanatory power was substantial ($R^2 = 0.33$) and significantly better than a null model that assumed no effect of need or merit ($\chi^2$ (5, $N = 28$) = 239.68, p < 0.001). Partners' merit ($\chi^2$ (2, $N = 28$) = 45.68, p < 0.001) and need ($\chi^2$ (1, $N = 28$) = 16.79, p < 0.001) both influenced generosity in the altruism task. On average, participants were more generous to a partner in high (vs. low) need (beta = 0.40, 95% CI [0.21, 0.58], p < 0.001). They were also more generous to a high (vs. low) merit partner (beta = 0.60, 95% CI [0.42, 0.79], p < 0.001) and an unknown (vs. low) merit partner (beta = 0.52, 95% CI [0.33, 0.71], p < 0.001). No interactions between the level of merit and need were significant ($\chi^2$ (2, $N = 28$) = 2.77, p = 0.25; need [high] × merit [unknown]: beta = 0.19, 95% CI [–0.06, 0.45], p = 0.142; need [high] × merit [high]: beta = 0.19, 95% CI [–0.07, 0.45], p = 0.147). Likewise, results of a formal model comparison revealed that adding the interaction of need and merit did not improve model fit significantly over a model that only considered the main effects ($\chi^2$ (2, $N = 28$) = 2.75, p = 0.252; *Supplementary file 4*). These findings suggest that need and merit inferences had fully independent effects on social choice. Consistent with this notion, we also found that merit-induced changes in generosity (high–low merit partner) and need-induced changes in generosity (high–low need) were uncorrelated (Spearman's $r = 0.17$, p = 0.377). In other words, people who changed their generous behavior as a function of another's merit were not necessarily the same as those who changed their behavior in response to another person's need. Consequently, further analyses focused on the main effects of need and merit on altruistic choice. Overall, the experimental manipulations of another's need and merit affected people's generosity during altruistic choice.

## Computational behavioral model of altruistic choice: partners' merit and need alter social attribute weights

Several mechanisms might drive the changes in altruistic choices we observed (see above): others' merit or need might decrease self-interest, increase other-regard, increase fairness considerations, or some combination of these. To identify the relative contribution of the processes that drive generosity, we turned toward an established behavioral computational model of altruistic choice (*Equation 4*; see Methods). We first verified that our computational model of altruistic choice fit the data well by showing that choices and reaction times (RTs) were captured with high accuracy (for visualization of model fit, see *Figure 5—figure supplement 1*).

Next, we examined the relative importance of the social attributes (drift weights *w*), irrespective of the partner's need or merit. On average (across all conditions), self-related outcomes strongly guided choices in the altruism task (indicated by the average overall positive weight on payoffs for self, $w_{self} = 0.97 \pm 0.75$, mean ± std), more so than concern for others' outcomes (overall $w_{other} = 0.22 \pm 0.49$, p < 0.001; FDR corrected) or fairness (overall $w_{fairness} = 0.35 \pm 0.36$, p = 0.028; FDR corrected). These results match the findings of prior studies (*Tusche and Hutcherson, 2018*; *Hutcherson and Tusche, 2022*). Variance in these *overall* attribute weights across people served as an indicator of more general, context-independent differences in (pro)social decision-making (explaining who will be more generous, on average, and why).

Finally, we tested if contextual cues about the partner's merit or need altered the degree to which benefits for oneself ($w_{self}$), others ($w_{other}$), or fairness concerns ($w_{fairness}$) guided social choices in the altruism task (separate Wilcoxon signed rank tests for each attribute and experimental manipulation; FDR corrected for multiple comparisons; see *Supplementary file 5* for attribute-specific estimates; see *Figure 5B, C* for illustration). Any changes in model estimates would indicate *context-dependent* effects (of varying levels of merit and need) of attributes' input on altruistic behaviors.

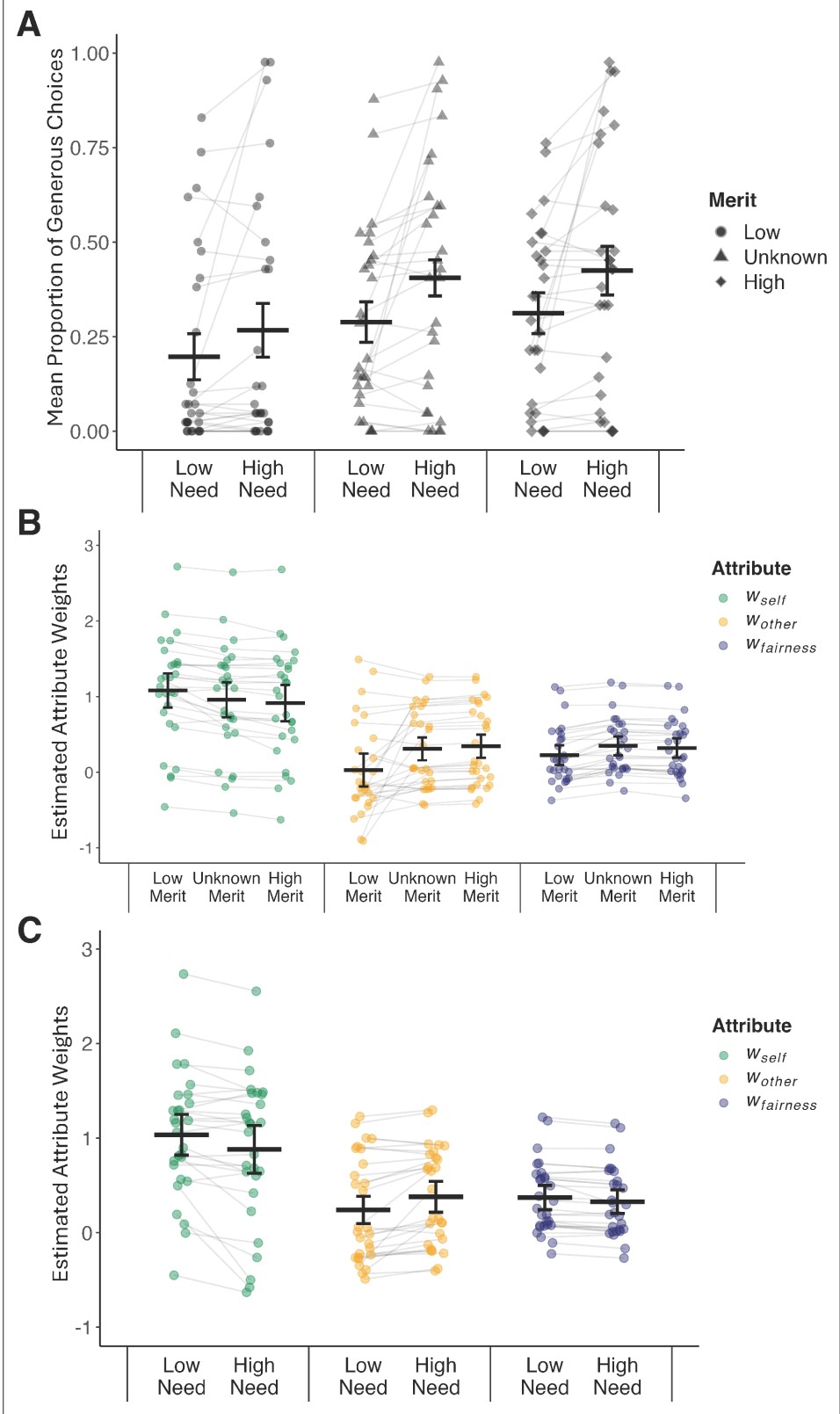

**Figure 5.** Effects of partner's merit and need on altruistic decision-making. (**A**) Partner's merit and need altered generosity in the altruism task. High (vs. low) need contexts elicited more generosity (p < 0.001). Compared to a low merit partner (circle), generosity was enhanced toward a high merit (diamond, p < 0.001) and an unknown merit partner (triangle, p < 0.001). (**B**) Condition-specific attribute weights ($w_{self}$, $w_{other}$, $w_{fairness}$) for low, unknown, and

*Figure 5 continued on next page*

*Figure 5 continued*

high merit partner contexts. (**C**) Condition-specific attribute weights for low and high need contexts. All p's ≤ 0.01, FDR corrected, for the six pair-wise comparisons of changes in attribute weights (high vs. low merit/need). Dots represent participant-specific estimates from the computational model of altruistic choice; black lines illustrate the estimated means and 95% confidence intervals.

The online version of this article includes the following figure supplement(s) for figure 5:

**Figure supplement 1.** Model fit of our computational model of altruistic choice illustrated via the correspondence between predicted (model) and observed (data) choices (left panel) and reaction times (RTs; middle and right panel).

When interacting with a high (vs. low) merit partner, benefits for oneself ($w_{self}$) guided choices *less* (p < 0.001, FDR corrected), whereas considerations of others' benefits ($w_{other}$, p < 0.01, FDR corrected) and fairness concerns (overall $w_{fairness}$, p < 0.001, FDR corrected) guided choices *more*. Mirroring this pattern of results, in high (vs. low) need settings, weights on self-related outcomes *decreased* (p < 0.01, FDR corrected), whereas weights on others' benefits *increased* (p < 0.001, FDR corrected). Unlike the merit-induced effects, a partner's high (vs. low) need *reduced* weights on fairness concerns on choices (p < 0.001, FDR corrected). Thus, if others faced great need, participants were more willing to ignore their fairness preferences. Together, these results suggest that merit- and need-evoked changes in generosity are driven by systematic changes in the social decision process—namely changes in attribute weights—as captured in our behavioral computational model of altruistic choice. Thus, we addressed another key question by showing that partners' merit and need levels changed how specific choice-relevant considerations (self-regard, other-regard, and fairness) guided people's decisions to act prosocially.

## Individual differences in effects of others' need and merit and altruistic decision-making

Notably, people differed substantially in their overall generosity and the degree to which generosity varied as a function of their partner's need and merit (*Table 2*). Our model assumes this variation was driven by changes in social attribute weights *w* estimated in the behavioral computational model (see above). To quantify these individual differences and examine their relations, we calculated change scores of model-based estimates for each participant. For example, to capture the change in an individual's other-regard as a function of their partner's need, we computed the following participant-specific difference score [$w_{other}$ in high need − $w_{other}$ in low need]. Likewise, to assess the change in

**Table 3.** Correlations between changes in generosity and changes in parameter estimates of the computational model of altruistic choice (attribute weights) across conditions in the altruism task.

| Condition Altruism task | Changes in parameter estimates | Merit-induced changes in generosity [High merit − Low merit] | | Need-induced changes in generosity [High need − Low need] | |
|---|---|---|---|---|---|
| | | *R* coefficient | p-value | *R* coefficient | p-value |
| Merit [High–low] | | | | | |
| | $\Delta w_{self}$ | −0.85 | 0.000* | −0.19 | 344 |
| | $\Delta w_{other}$ | 0.91 | 0.000* | 0.01 | 0.948 |
| | $\Delta w_{fairness}$ | 0.56 | 0.018* | −0.11 | 0.569 |
| Need [High–low] | | | | | |
| | $\Delta w_{self}$ | 0.19 | 0.333 | 0.75 | 0.000* |
| | $\Delta w_{other}$ | −0.14 | 0.484 | −0.88 | 0.000* |
| | $\Delta w_{fairness}$ | −0.11 | 0.570 | −0.03 | 0.895 |

Note. Spearman's correlation coefficients, FDR corrected. * Indicates significant correlations.

other-regard in response to their partner's merit, we estimated the change in [$w_{other}$ high merit − $w_{other}$ low merit]. We did this separately for each attribute weight estimated in the computational model of altruistic choices ($w_{self}$, $w_{other}$, $w_{fairness}$). These change scores reflect the impact of partners' merit and need on individuals' altruistic decision process.

As a sanity check, we confirmed that changes in attribute weights reflect participant-specific changes in generosity (*Table 3*). In other words, larger (smaller) changes in observed social behaviors can be explained by larger (smaller) changes in estimated attribute weights. Specifically, we confirmed that individuals' merit-induced changes in generosity were reflected by individuals' altered weights for self-interest (payoffs for self), other-regard (payoffs for partners) and fairness considerations on choices (p's < 0.018, FDR corrected). Need-related changes in generosity were reflected by altered consideration of benefits for self and others (p's < 0.001, FDR corrected; but not fairness, p = 0.895). In other words, individuals who were most sensitive to information about others' merit and need (i.e., large differences in generosity toward a high vs. low merit/need partner) changed their decision process more strongly (captured in larger shifts in the weights on outcomes for self, other, and fairness considerations on choices as a function of others' merit/need). These findings provide insights into the precise mechanism by which need and merit affect (pro)social behavior in altruistic choice settings, namely by altering the weighting of certain choice attributes in the decision-making process. Below, we link these change scores in social behaviors with estimates of social perception (sensitivity, bias).

## Variance in perceptual sensitivity and bias (social perception task) predict variance in prosocial behavior across people and contexts (altruism task)

The previous section contained three main takeaways regarding the separate altruism task. First, partners' merit and need independently impacted social behaviors (generosity). Second, we identified the mechanism of these context-dependent changes in social behaviors: merit and need altered the relative importance of self- and other-regard, and fairness preferences (captured in attribute weights *w* estimated in our computational model of altruistic choice). Third, we demonstrated that people differ in the degree to which they change their behaviors in response to social cues on others' merit or need. This raises an important question: what factors determine the impact of others' need or merit on behavior?

We propose that stable individual differences in social perceptions—as captured in our computational model of social perception and their neural underpinnings—can provide insights into this question. Behaviorally, the computational model decomposes individual differences in social perceptions into bias and sensitivity terms. These model estimates from the social perception task correspond to two mechanisms that drive variation in people's social perceptions of others' merit and need. We propose that these stable perceptual mechanisms can, in turn, impact social decision-making. Here, we bring together data from both tasks: the social perception task (which did not require meaningful social behavior toward others) and the altruism task (which included costly social actions). We had two specific hypotheses in mind when designing the study: we speculated that merit bias (or need bias) in the social perception task should be related to *average* weights on others' outcomes in the altruism task. In contrast, estimates of an individual's sensitivity to merit (or need) should predict the extent to which a person *alters* their weight on others' outcomes as a function of their partner's merit (or need). By combining data from both tasks, we explore a fundamental question: do people's sensitivities and biases during social perceptions translate into subsequent social action? Notably, we see the results below as evidence of *stable* individual differences, since social action (i.e., generosity) was measured on average almost 10 months (~303 days) after the social perception task.

## Perceptual merit sensitivity predicts merit-related contextual changes in altruistic choice

Estimates from our computational behavioral model map onto specific hypotheses about the relationship between behaviors observed in both tasks. Perceptual sensitivity estimates reflect individuals' tendency to sample and integrate evidence about merit (need) during the perceptual decision process. Higher sensitivity estimates will yield higher discriminability between others as a function of their merit (need). Assuming stable individual differences in sensitivity and the impact of categorizing individuals as meritorious or needy on altruistic decision-making, we hypothesized that individuals

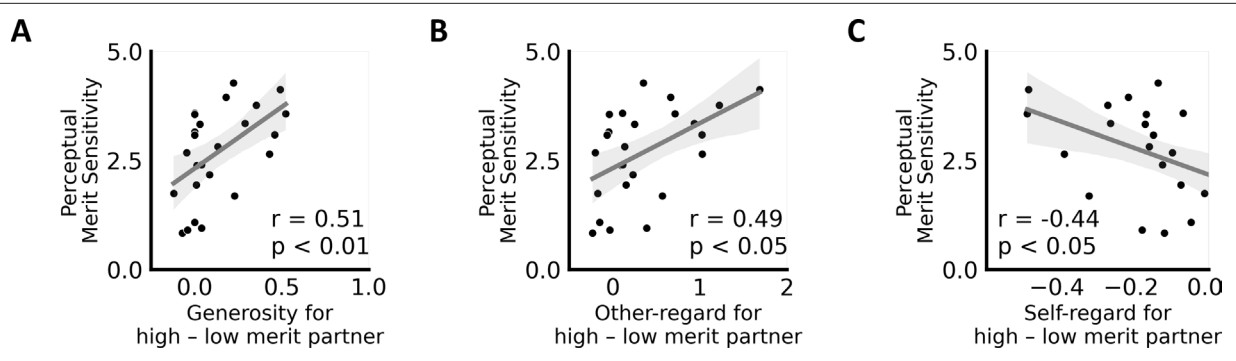

**Figure 6.** Individual differences in merit sensitivity predict social preferences in altruistic decisions. (**A**) Variance in individuals' general sensitivity to others' merit (social perception task) is positively linked with merit-related changes in generosity, (**B**) other-regard ($w_{other}$), and (**C**) self-regard ($w_{self}$) during altruistic choice for high versus low merit partners (altruism task). Higher values on the *x*-axes indicate increased generosity/other-regard and decreased self-interest when interacting with a high versus low merit partner in the altruism task.

with higher perceptual merit (need) sensitivity in the social perception task would exhibit a greater change in social behavior depending on the partner's merit (need) during the altruism task. To test this hypothesis, we correlated individuals' merit sensitivity scores (social perception task) with merit-induced changes in social behavior (i.e., participant-specific changes in generosity toward [high − low merit] partners in the altruism task). We found that variance in merit sensitivity reflected merit-induced changes in generosity (Spearman's *r* = 0.51, p = 0.010; *Figure 6A*). In other words, individuals generally sensitive to merit information during social perceptions were also more susceptible to merit information during costly social behaviors. Follow-up analyses using estimates of a computational model of the altruism task revealed that this perception–action link was driven by merit-induced changes in other- and self-regard during altruistic choice (change in $w_{other}$ for [high − low merit] partners: Spearman's *r* = 0.49, p = 0.045, FDR corrected; change in $w_{self}$ for [high − low merit] partners: Spearman's *r* = –0.44, p = 0.047, FDR corrected; no significant link with merit-related changes in fairness weight, p = 0.784, FDR corrected; *Figure 6B, C*). In other words, individuals with higher merit sensitivity showed larger discrimination in the value placed on others' wellbeing (and self-interest) when interacting with supposedly deserving and undeserving people. Next, we repeated this set of analyses for sensitivity scores estimated in the need condition of the social perception task. Here, variance in participants' need sensitivity did not reflect need-induced changes in generosity (p = 0.745) or need-induced changes in altruistic choice attributes (all p's > 0.484). However, this absence of effects for need ought to be interpreted with caution, given the comparatively small sample size. Results were qualitatively similar when statistically controlling for the delay between both tasks (partial correlations).

### Perceptual bias estimates predict individuals' overall other-regard, self-regard, and fairness considerations in altruistic choice

We also hypothesized that people's stable perceptual biases in the social perception task might translate into *context-independent* differences in social action (generosity) across people. Put differently, we assumed that individuals who have a *general* tendency to perceive others as deserving/in need (irrespective of social cues present in the environment) should be more willing to help others irrespective of contextual variation in others' merit/need. We tested this notion by correlating merit and need *bias* parameters (social perception task) with individuals' overall generosity and the overall weight of choice-relevant attributes in the altruism task (i.e., outcomes for self, $w_{self}$, outcome for other, $w_{other}$, and fairness concerns, $w_{fairness}$). Contrary to our hypothesis, individuals' merit bias scores were not correlated with overall generosity (p = 0.282). However, we did find that people who tend to perceive others as deserving tended to be more other-oriented overall during altruistic choices: variance in merit bias scores in the social perception task positively correlated with the weights on others' benefits (*overall* $w_{other}$, Spearman's *r* = 0.50, p = 0.035, FDR corrected; marginal positive link with overall fairness concerns in the altruism task, *overall* $w_{fairness}$, Spearman's *r* = 0.43, p = 0.053, FDR corrected; no significant link was observed with overall self-regard, p = 0.153, FDR corrected). This finding suggests that stable differences in people's tendency to perceive others as deserving predicts

people's overall other-regard across different social choice contexts (on average) 10 months later. In contrast, variance in individuals' need bias (social perception task) in our sample did not correlate with overall generosity (p = 0.150) or overall attribute weights (all p's > 0.076, FDR corrected). As mentioned above, we cannot rule out the possibility that null findings may be due to the comparatively small sample size and should be interpreted cautiously (also see discussion). Results were qualitatively similar when statistically controlling for the delay between both tasks (partial correlations).

## Neural markers of merit sensitivity predict merit-related behavioral changes during altruistic choice

So far, within the social perception task, we found that merit-evoked neural activation in the rTPJ reflects variance in people's merit sensitivity (*Figure 3B*). Moreover, when combining data from both tasks, we showed that individuals' merit sensitivity (social perception task) predicts merit-related changes in other- and self-regard that guide context-dependent changes in social behavior (altruism task) (*Figure 6B, C*). Considering these findings, a post hoc test examined whether activity in the rTPJ—obtained during merit inferences (social perception task) —also predicts merit-related changes in other-regard (or self-regard) in the altruism task (beyond merit sensitivity). We tested this question by using the following equation:

$$\Delta w_{other\,j} \sim 1 + S_{merit\,j} + rTPJ_j \qquad (2)$$

Here, $\Delta w_{other\,j}$ represents an individual $j$'s merit-related change in other-regard during altruistic choice ($w_{other}$ for high merit − low merit partners). We used two predictor variables: participants' behavioral merit sensitivity scores ($S_{merit\,j}$; estimated in the computational model of social perception) and participant-specific neural responses in the rTPJ obtained during merit inferences (merit − control) in the social perception task (all voxels of the rTPJ cluster, averaged across all voxels in the cluster, see *Table 1*). This allowed us to assess the additional predictive power of neural responses in the rTPJ, after controlling for behavioral merit sensitivity. The model's total explanatory power was substantial ($R^2 = 0.40$) and significantly better than a null model with just an intercept ($\chi^2$ (2, $N = 25$) = 2.46, p < 0.001). We found that both model-based estimates of individuals' merit sensitivity and rTPJ responses reliably, independently, and to an equivalent degree predict changes in other-regard during social choice approximately 10 months later ($S_{merit}$: beta = 0.20, SE = 0.08, 95% CI [0.04, 0.36], $t(22) = 2.50$, p = 0.021; rTPJ: beta = 0.29, SE = 0.13, 95% CI [0.05, 0.54], $t(22) = 2.34$, p = 0.029). These findings indicate that neural correlates of merit inferences—namely activity in the rTPJ—predict context-dependent changes in other-regard during social action across time and contexts, above and beyond predictive information related to perceptual merit sensitivity.

For completeness, we also estimated a modified version of *Equation 2* in which we changed the dependent variables to merit-evoked changes in self-regard in the altruism task (replacing $\Delta w_{other\,j}$ with $\Delta w_{self\,j}$, defined as participant-specific change scores in self-regard when interacting with a high vs. low merit partner). The model yielded an $R^2$ of 0.37 and the rTPJ remained a significant predictor (beta = –0.09, 95% CI [–0.17, –0.02], $t(21) = -2.47$, p = 0.022). However, the behavioral merit sensitivity was reduced to only marginal significance ($S_{merit}$: beta = –0.05, 95% CI [–0.10, 0.00], $t(21) = -1.92$, p = 0.069). Thus, rTPJ responses during merit inferences were tied to estimates of contextual changes in both other- and self-regard to an equivalent or possibly even greater degree than behavior alone.

## Discussion

Humans do not help others indiscriminately: they are more inclined to help people perceived as needy or deserving (*Molouki and Bartels, 2020*; *Hare et al., 2010*; *Tusche et al., 2016*; *Fong, 2007*; *Eckel and Grossman, 1996*; *Engel, 2011*; *Jilke and Tummers, 2018*). Using a novel fMRI social perception task, we disentangled two distinct computational mechanisms that shape variance in these social judgments: a general *bias* to perceive others as more or less deserving (in need) and a degree of discrimination or *sensitivity* to social cues signaling others' merit (need). Estimates of these two computations were uncorrelated, suggesting they represent distinct—but not mutually exclusive—processes driving individual differences in perceptions of peoples' merit and need. We also demonstrated that these computations (for merit, if not need) might be stable and generalizable over time: individuals' perceptual merit sensitivity predicted the degree to which they discriminated

between others based on merit in a separate altruism task completed from 27 to 663 days later. Moreover, their perceptual merit bias predicted a general propensity to weigh others' outcomes instead of their own during altruistic choices. Neurally, merit sensitivity (but not bias) was associated with increased activity of the TPJ during perceptual judgments, which in turn predicted merit-related discrimination in altruistic behavior. Together, our results identify a set of distinct neurocomputational mechanisms that contribute to our understanding of *when* and *how* perceptions of others translate into social actions.

## Translating perception into action

Variance in people's sensitivity in merit perceptions predicts context-specific social behaviors and discrimination. This finding contributes to a growing literature regarding parochial altruism. Parochial altruism refers to the tendency to exhibit altruistic behavior toward in-group members and to withhold it from (or even display hostility toward) out-group members (*Bernhard et al., 2006*). Parochial altruism occurs around the world in private and public settings (*Romano et al., 2021*), in sports, politics, war, and religion (*Hein et al., 2010*; *Ginges et al., 2009*; *Choi and Bowles, 2007*; *Brewer et al., 2023*), and has also been linked to activation patterns in the TPJ (*Baumgartner et al., 2012*; *Obeso et al., 2018*; *Fujino et al., 2020*). Although we did not use group- or membership-based cues to characterize partners in the altruism task (manipulating merit instead via partner *behaviors* in a separate task), the common locus in the TPJ might indicate shared mechanisms for both types of context-dependent social behaviors. Interestingly, our results suggest a considerable degree of stability of this idiosyncratic perceptual sensitivity across time and contexts, since the social perception and altruism tasks were completed on average ~10 months apart, up to almost 2 years for some participants. This finding is consistent with research showing a hereditary component of in-group favoritism and parochial altruism (*Lewis and Bates, 2010*). Future work should examine the extent to which merit-related perceptual sensitivity represents an innate/genetic or learned quantity, and how it correlates with other types of discriminatory behaviors. Our results also speak to the literature on universal altruism: individual differences in the *bias* to perceive merit correlated with the *overall* weight an individual placed on others during altruistic choice months later. This finding supports empirical evidence that dispositionally cooperative people are more universal in their cooperation (*Romano et al., 2017*) and suggests that this could partly come from a generalized bias to perceive people as deserving. Although we did not find a distinct neural signature of this bias, future work on genetic or anatomical differences might yield clearer results.

## The value of computational decomposition of perception and action

The power of formal computational models to uncover patterns, principles, and dynamics in social perception and behavior (*Crockett, 2016*; *Johnson et al., 2017*; *Hackel and Amodio, 2018*; *Freeman et al., 2018*) has made them an increasingly popular tool in economics, psychology, and neuroscience (*Capraro and Perc, 2021*; *Kvaran and Sanfey, 2010*; *Kliemann and Adolphs, 2018*; *Charpentier and O'Doherty, 2018*; *Behrens et al., 2009*; *Freeman and Ambady, 2011*; *Yu et al., 2019*). Our results contribute to this movement by showing how computational models of social perception can provide novel insights into the different computations (i.e., bias and sensitivity) underlying impression formation and its effect on behavior. Although the concept that judgments are a composite of subprocesses is not novel in itself (*Kenny and Albright, 1987*), modeling allowed us to formally disentangle these different (neuro)computational mechanisms and demonstrate how they shape different aspects of meaningful social action months (or even years) later. Future research should confirm and extend these findings using other social judgments in other, ideally more ecologically relevant, contexts (*Thornton and Mitchell, 2018*). For example, research suggests that perceptions of others' warmth and competence can impact hiring decisions (*Cuddy et al., 2011*; *Louvet, 2007*). Are perceptions of warmth or competence likewise driven by stable individual differences in bias and sensitivity? Do stable individual differences in merit bias or sensitivity shape real-world prosociality, such as political support for social welfare programs (*Appelbaum, 2002*; *Attewell, 2021*; *Oorschot Van, 2000*), or the extent to which different people engage in the online posting of degrading content, harsh comments, or cyber-bullying (*Coe et al., 2014*; *Tokunaga, 2010*)?

### The neural bases of need and merit perception

Determining whether socially relevant cues are processed by domain-specific or domain-general neural circuitry remains an active goal of the social and cognitive neurosciences. While considerable evidence exists for specificity in some domains (e.g., face processing *Elbich et al., 2019*; *Itier et al., 2007*; *Kanwisher, 2000*), emotion recognition (*Ekman et al., 1983*; *Adolphs et al., 1994*; *Morris et al., 1996*), empathy in different modalities (*Corradi-Dell'Acqua et al., 2016*), or even aspects of moral decision-making (*Hutcherson et al., 2015b*), other research points to the broad engagement of the mentalizing network across tasks (*Barrett and Satpute, 2013*; *Schuwerk et al., 2017*; *Scholz et al., 2009*; *Mitchell, 2008*; *Baetens et al., 2015*; *Carter and Huettel, 2013*; *Lugrin et al., 2023*). Given their importance for decision-making, we sought to determine whether perceptions of either need or merit fall within the category of social phenomena processed by dedicated neural circuits. Our neural findings largely suggest the answer is no. Both merit and need perception engaged the mentalizing network to a similar extent and were virtually indistinguishable neurally, with some minor differences. This held true even when applying multivariate decoding approaches, which have been suggested to be more sensitive than traditional univariate analysis techniques (*Kriegeskorte and Bandettini, 2007*). We note, however, that perceptions of others' merit/need in this task likely represent a composite of multiple different sub-components (e.g., related to specific social cues sampled and integrated to yield the final social judgment). Thus, our results do not preclude the existence of domain-specific neural circuitry at a lower level of social appraisals (e.g., gender or age categorization, facial and postural cues that a person is in pain, etc.).

### Limitations and future directions

Our results come with some important limitations. One of the biggest concerns is the puzzling absence of any observed neural or behavioral correlates of need sensitivity, in either the social perception or altruism tasks. We primarily relied on univariate analyses to support our conclusions here. Although a supplementary multivariate pattern analysis yielded little additional insight, exploring alternative methods such as the gradient approach or functional connectivity (*Janet et al., 2024*; *Margulies et al., 2016*; *Vos de Wael et al., 2020*) could prove more revealing. One other contributing factor to this null result could simply be the small sample size for our altruism task due to COVID-related delays and participant attrition. Other alternative explanations are also possible, however. While purely speculative, one possibility might lie in a limitation of the stimuli used for the social perception task. Need was generally signaled directly and concretely in each picture (i.e., someone with a knife to their throat, someone with a pained facial expression, etc.). In contrast, merit often had to be inferred indirectly from cues about the cognitive and motivational dispositions of the target (e.g., performance of unethical actions, clothing indicating group membership, other signs this person was a 'good' person, etc.). The ability to discriminate based on these abstract cues might thus have correlated with TPJ response and with the use of similarly abstract cues during the altruism task to judge merit of social partners. It is possible that if we had signaled need using similarly abstract cues in the social perception task, we might have observed greater associations with mentalizing network regions and/or with altruistic behavior months later. Future work will be needed to more systematically vary factors related to concrete versus abstract inference, in larger and more diverse samples, and with a greater range of socially relevant behaviors. Finally, another important open question concerns the origins of the identified perceptual biases and sensitivities. Future research should examine if differences in social perceptions stem from societal or religious values (see *Amormino et al., 2022*) conveyed during individuals' upbringing, genetic factors, or a combination of both.

## Materials and methods
### Participants

This research took place in the context of a large project that recruited participants from the broader Los Angeles metropolitan area to come to the lab on multiple days to complete different behavioral and neuroimaging tasks related to social cognition (*Kliemann et al., 2022*). We recruited 50 participants from this larger project pool to complete a newly developed fMRI social perception task. All participants were right handed, had normal or corrected-to-normal vision, spoke English fluently, and had IQ scores in the normal range (20 females; mean age = 33 years, range = 19–49; full-scale

IQ = 105.18 ± 8.04 (mean ± std), range = 87–127; *Wechsler, 1999*; *Wechsler, 2011*). Of these 50 participants, we excluded one who fell asleep and five for excessive movement (framewise displacement >0.3 mm on over 30% of frames and visual spikes), yielding a total sample of $n$ = 44 for the social perception task. We also recruited 42 participants from the larger participant pool to complete the altruism task (13 females; mean age = 34 years, range = 19–49; full-scale IQ = 104.79 ± 7.97, range = 87–127). Of these 42 participants, we excluded one individual who fell asleep, five for excessive movement during fMRI data collection, three for invariant behavioral responses (identical left/right button press in >90% of trials, indicating inattention to monetary offers in the altruism task), and five for manipulation check failures (e.g., failing memory checks regarding partner behavior in the altruism task, see below). This procedure yielded a sample of $n$ = 28 for the altruism task, with an overlap of 25 individuals who successfully completed both sessions (social perception and altruism tasks). Thus, we report results regarding the social perception task for 44 participants (27 males; mean age = 34 years, range = 19–49; full-scale IQ = 105.84 ± 7.85, range = 90–127), the altruism task for 28 participants (19 males; mean age = 35 years, range = 25–49; full-scale IQ = 106.36 ± 7.94, range = 94–127), and cross-session results (comparing data across both tasks) for the sample of 25 individuals with valid data in both (17 males; mean age = 35 years, range = 25–49; full-scale IQ = 106.60 ± 8.26, range = 94–127). Participants in our overlap sample (valid data in both tasks) had an average separation between both tasks of 303 days (range: 27–663). This delay minimized the risk of sequential dependencies between tasks and increased confidence in the temporal stability of relationships between social perception and social action. Participants received $20 per hour for each experimental session and an additional amount based on one randomly selected trial in the altruism task to incentivize choices and ensure that participants' responses reflected their actual preferences. All participants provided written informed consent according to a protocol approved by the Institutional Review Board of the California Institute of Technology (#12-0343).

## Normative sample

Estimating participants' sensitivity to need and merit cues in the social perception task required quantifying the degree of need and merit displayed in each picture (e.g., stimuli, see *Figure 1A*). We assessed this quantity by obtaining normative ratings from an independent sample of participants recruited through Mechanical Turk (MTurk; http://www.mturk.com) and Qualtrics (http://www.qualtrics.com). The normative sample included 50 participants (17 females; mean age = 42 years, range = 25–67; 82% of White race; 100% native English speakers). This sample performed an online behavioral version of the social perception task (see below). Participants made binary judgments of whether the displayed target individual needed help, deserved help (merit), or used both hands (control) (i.e., the same judgment made by our main fMRI sample). For a given judgment (need/merit/control), the mean proportions of 'yes' responses across the sample for each image were used to operationalize perceptual evidence on merit, need, and control in the experimental stimuli used in the social perception task. We used these data from the separate sample to estimate the free sensitivity parameters in the behavioral computational model of social perception (see below). Data from our independent normative participant sample are available on the Open Science Framework (OSF; see https://osf.io/4u5vs/).

## Social perception task

To assess individual differences in social perceptions, we used a modified version of the established fMRI why/how task (*Spunt and Adolphs, 2014*; *Spunt and Adolphs, 2015*). Participants viewed images of people in complex real-world scenes and made rapid yes/no judgments (button presses) while their brain responses were measured using fMRI. In separate blocks (*Figure 1A*), participants made judgments regarding others' perceived merit (Does this person deserve help?), need (Does this person need help?), or factual judgments that did not require social inferences (Does this person use both hands?), which served as a control condition. Before each task block, a visual prompt informed participants of the upcoming condition of the task block. Moreover, a keyword was briefly presented between images of a block as a reminder (i.e., 'need', 'deserve', or 'both hands'; *Figure 1A*). Each block consisted of 32 images, and participants completed two blocks per condition. Thus, participants viewed a total of 64 images, each presented once per condition (need, merit, and control). Some images displayed multiple people; a green arrow superimposed on the photograph indicated the target of the social perception. Images were displayed for 2 s with a 0.5 s inter-stimulus interval. The

presentation order of images was fixed across participants to maximize efficiency. Crucially, this task did not require people to make altruistic choices or engage in meaningful social interaction. Instead, it focused solely on capturing individuals' patterns of perceiving others' perceived merit, need, and factual control inferences, which we modeled using an evidence accumulation framework (computational model of social perception, see below). The task was implemented in MATLAB (MathWorks) using the Psychophysics Toolbox extensions (*Brainard, 1997*; *Kleiner et al., 2007*; *Pelli, 1997*). The stimuli and presentation code are available at https://osf.io/4u5vs/.

## Altruism task

Participants completed an altruism task (modified dictator game) on a different day than the social perception task (average delay of 303 days, min. = 27, max. = 663). All but one participant completed the social perception task first. On each trial, we presented participants with a proposed monetary allocation between themselves and one of three partners (e.g., $13 for themselves and $35 for their current partner; choice screen, see *Figure 4A*). Participants decided whether to accept or reject the proposed allocation in favor of a constant default allocation of $20 for both (*Hutcherson et al., 2015a*; *Tusche and Hutcherson, 2018*). Participants indicated their choices by pressing one of four buttons ('strong no', 'no', 'yes', or 'strong yes'). The direction of the response scale ('strong yes' to 'strong no') and the presentation side of self- and other-related payoffs (left vs. right side of the screen) was counterbalanced across participants. Proposed monetary outcomes for the participant (You) and the partner (represented by one of three colored geometric shapes) ranged from $5 to $35 (*Figure 4B*). To minimize the effects of habituation and repetition, we randomly jittered proposal amounts by $0–$4. We informed participants that one trial would be randomly selected and implemented at the end of the experiment. In total, there were 300 trials divided across five runs (i.e., 60 trials per run, 100 trials per partner). Stimulus presentation and response collection in the altruism task were implemented using PsychoPy (*Peirce, 2007*; *Peirce, 2008*).

### Partner's merit (high/low/unknown)

Participants played the altruism task with three partners that differed in their perceived merit (implemented in separate blocks; 20 consecutive trials per partner within a functional run; counterbalanced order of the three partners across participants). The merit of the three partners in the task was manipulated before the altruism task and was based on information about partners' behavior in a separate exchange game that partners played with anonymous third persons, modified from *Singer et al., 2006*; *Singer et al., 2004* (see Appendix 1). Merit levels were manipulated such that one player was perceived as highly deserving (high merit partner), undeserving (low merit partner), or having unknown merit (no information provided before the altruism task, control condition). Partner identity was indicated in the altruism task using one of three colored geometric shapes (random combination of either a circle, diamond, or triangle, colored either red, yellow, or blue), shown at the beginning of each block of 20 trials with the same partner (merit cue, *Figure 4A*) and on each trial-wise offer screen indicating payoffs for the specific partner (choice screen, *Figure 4A*).

### Partner's need (high/low)

The altruism task also experimentally manipulated the need level of the three partners on a trial-by-trial level. Specifically, we manipulated the probability that a partner would have to complete a painful cold pressor task (CPT; *Hines and Brown, 1936*) at the end of the experiment (outside of the scanner; hand submerged in ice water for ~2 min). We informed participants that each of the three partners would be given the option of using money received in the altruism task to buy out of the post-task CPT (based on a randomly selected trial at the end of the task that would be implemented according to the participants' choice on that trial). With each dollar spent, the partner could subtract 10% from their probability of having to perform the painful CPT (e.g., spending $3 would reduce an 80% chance to 50%). Thus, participants knew they could help the other player avoid the painful CPT by making generous choices.

To signal need on each trial, participants were presented with a percentage indicating the others' need (high vs. low probability of CPT) before seeing the proposed monetary allocation (need cue, *Figure 4A*). Thus, in trials with high percentages, the partners had a greater need for money to avoid the painful experience. For each partner, CPT likelihood was low in half of the trials (mean CPT

probability of 20%) and high in the other half (mean CPT probability of 80%; random presentation order of high vs. low need trials). As with the monetary proposal amounts, we randomly jittered need-related percentages by 0–4% to minimize the effects of habituation and repetition.

To ensure the saliency of the experimental need manipulation, all participants completed the painful CPT task themselves before the altruism task outside the scanner. Participants were also told that all three partners had already completed one round of the CPT task before the altruism task and were presented with their ostensible pain ratings (7 on a 7-point scale; 1 = not at all; 7 = extremely). These ratings signaled to participants that all three partners found the ice water equally and extremely painful and were highly motivated to avoid another round of the CPT. After the altruism task, participants completed a variety of computerized sanity check questions and questionnaires outside of the scanner (see *Figure 4—figure supplement 1*). These sanity checks were used to verify the effectiveness of our experimental merit and need manipulations and to exclude any participants who failed to correctly remember how partners had acted in the behavioral exchange game prior to the altruism task (Appendix 1).

## Analysis
### Computational behavioral model of social perception
We developed two behavioral computational models to characterize individual differences in social perceptions (social perception task) and prosocial behaviors (altruism task). Separately for each task, we modeled participants' choices and RTs using variants of the drift-diffusion model (DDM; *Ratcliff et al., 2016*). This model depicts choices as the noisy accumulation of evidence until a sufficient level favoring one choice option is attained. The DDM has been used to examine processes underlying both perceptual and value-based decisions (*Ratcliff et al., 2016*; *Krajbich et al., 2010*; *Krajbich and Rangel, 2011*; *Towal et al., 2013*) and is being increasingly applied to studying social and affective decision-making processes (*Tusche and Bas, 2021*; *Saulin et al., 2022*; *Hutcherson et al., 2015a*; *Roberts and Hutcherson, 2019*; *Krajbich et al., 2015*; *Kutlikova et al., 2023*; *Harris et al., 2018*; *Pollerhoff et al., 2023*).

To model trial-wise responses in the social perception task (*Figure 1A*), we assumed that when faced with the task of judging each image on a particular dimension (i.e., merit, need, or the control judgments of using both hands), a decision-maker employs the following strategy. At each moment in time, they draw noisy samples of both task-relevant and task-irrelevant evidence about *Merit*, *Need*, and *Control* (both hands) from the stimuli, weighted by person- and condition-specific sensitivities $S_{merit}$, $S_{need}$, and $S_{control}$. They accumulate these samples of evidence $E$ at each timepoint $t$ according to the following equation:

$$E(t|Condition) = Bias_{condition} + S_{merit|condition}Merit + S_{need|condition}Need$$
$$+ S_{control|condition}Control + \epsilon(t)$$

(3)

*Condition* refers to the specific judgment (merit, need, and control) being performed on that trial (task block). *Equation 3* reflects the assumption that a person's judgment should be most sensitive to task-relevant information (e.g., cues signaling merit during the merit judgment blocks), but might also be inadvertently influenced by task-irrelevant information (e.g., cues signaling need in merit judgment blocks). For example, merit judgments should primarily reflect cues indicating merit (i.e., high sensitivity estimates of $S_{merit|merit\ condition}$) but might also reflect in part cues indicative of need (i.e., a low but non-zero $S_{need|merit\ condition}$). Thus, the model yields a total of nine sensitivity parameters estimated across the three conditions of the social perceptual task. The image-level evidence for *Merit*, *Need*, and *Control* (for the stimulus shown on that trial) was estimated using data from an independent participant sample (see Normative sample above). For each image, we used the mean-centered average proportion of 'yes' responses that the displayed target individual deserved help, needed help, or was using both hands (based on data from the independent normative sample). We focus on the 'task-relevant' estimated parameters $S_{merit}$ in the merit condition, $S_{need}$ in the need condition, and $S_{control}$ in the control condition as indices of participants' perceptual *sensitivity* to cues suggestive of others' merit, need, or usage of both hands (supplemental manipulation checks confirmed larger perceptual sensitivity estimates in task-relevant compared to task-irrelevant task conditions, see Appendix 2). Higher parameter values suggest stronger discrimination of the normatively agreed-upon relevant cues (as captured by the independent normative sample). The DDM also includes three condition-wise

free parameters that influence the overall drift, irrespective of the specific image ($Bias_{need}$, $Bias_{merit}$, and $Bias_{control}$; indicated as $Bias_{condition}$ in *Equation 3*). These parameters allow us to capture an individual's general tendency to identify cues suggestive of others' merit and need (or control), regardless of the actual social cues present in the image. We refer to these estimates as perceptual *bias*.

Once the momentary evidence $E$ reaches the upper (yes) or lower (no) threshold, evidence accumulation terminates, and the corresponding choice is implemented. The difference between thresholds is estimated by a set of three parameters ($a_{\Delta need}$, $a_{\Delta merit}$, and $a_{\Delta control}$) that represent within-individual stability and difference across tasks. The DDM also includes three condition-wise non-decision time (*ndt*) parameters for capturing the time taken to initially encode stimuli and to implement the motor response, estimated similarly to incorporate within-individual stability and change across task conditions. Finally, the model included three condition-wise starting bias ($z$) parameters, which represent another potential mechanism by which biases at the onset of evidence accumulation can impact the decision process. Note that, although we estimated these starting biases to improve model fit, our focus in all primary analyses reported in this paper is on the evidence-related *perceptual* bias for merit and need (represented by $Bias_{condition}$ in merit and need blocks, respectively), not these motor-related starting biases (*Witt et al., 2015*). A detailed description of the model fitting procedure is provided in Appendix 3.

## Behavioral generosity in the altruism task

Choices in the altruism task (*Figure 4A*) involved a trade-off between monetary outcomes for the self and one of three partners. Monetary proposals were selected so that a choice could benefit the partner or the participant, compared to the constant default ($20 for both players). Following previous implementations (*Hutcherson et al., 2015a*; *Tusche and Hutcherson, 2018*; *Hutcherson and Tusche, 2022*), we classified a choice as generous if the participant *accepted* a proposal that benefited the partner at the expense of oneself ($Self < $Other) or *rejected* a proposal that profited themselves at the partner's expense ($Self > $Other). The overall and condition-specific fraction of generous choices measured participants' generosity. Differences in generosity across conditions provided model-free estimates of the impact of social cues about others' merit and need on social behaviors. To assess whether merit and need altered generosity, we computed a mixed-effects logistic regression using the *glmer* function in R. Trial-wise information about generous choice (no = 0, yes = 1) served as the dependent variable. The model included the following fixed effects: a trial-wise indicator for the level of the partners' need (reference: low need), an indicator for the partners' merit level (reference: merit low), and their interaction. Participant id was specified as a random effect, as follows:

$$\text{Generous Choice}(0/1) \sim 1 + \text{Need} + \text{Merit} + \text{Need} \times \text{Merit} + (1|\text{Participant}) \tag{4}$$

## Computational behavioral model of altruistic choice

To model choices and RT behavior in the 3 (merit: high, unknown, low) × 2 (need: high, low) design of the altruism task, we implemented a second behavioral computational model. Following previous applications (*Hutcherson et al., 2015a*; *Tusche and Hutcherson, 2018*), for each of the six experimental *conditions*, value-related evidence $V$ at time $t$ was estimated as follows:

$$V(t|condition) = w_{0|condition} + w_{self|condition}Self + w_{other|condition}Other$$
$$+ w_{fairness|condition}Fairness + \epsilon(t) \tag{5}$$

Here, *Self* and *Other* refer to the monetary outcomes of the proposed allocation of a trial (minus the default outcomes of $20 for both; rescaled by dividing by 10). The *Fairness* of the proposed allocation was calculated as $-1 \times |Other - Self|$. The free parameters for each attribute weight ($w_{self}$, $w_{other}$, $w_{fairness}$) indicate variance in the degree to which evidence about outcomes to the self, other, or the fairness of the monetary offer guided choices in a particular condition of the altruism task. The value constant $w_0$ represents the extent to which a participant tends to prefer the proposal over the default, regardless of trial-specific values. Matching the computational model of social perception (see above), the model of altruistic choice also estimated free parameters capturing the decision threshold ($a$), non-decision time (*ndt*), and response-related starting bias ($z$) parameter. The parameters $z$, *ndt*, and $a$ were fixed across the merit × need conditions.

To capture variance in weight parameters $w$ across the three levels of merit (high, low, unknown) and two levels of need (high, low), we estimated both individual-specific stability and change parameters for each of the four weights $w$ (i.e., self, other, fairness, and the value constant). A more complicated model that allowed for need × merit interactions in driving weights did not improve model fit. We thus focus on this simpler model that includes a baseline parameter for each weight, and three change parameters for the effects of increasing or decreasing merit and increasing need. A detailed description of the model fitting procedure is provided in Appendix 3.

### Correlating social perception and social action

Next, we tested for a relationship between social perception and social action. To this end, we correlated individual-specific bias and sensitivity parameters obtained from the social perception task with value-based attribute weight parameters in the altruism task. Because we found that some estimates of our computational model diverged from a normal distribution, we used non-parametric statistical tests to examine these relationships. Post hoc tests were corrected for multiple comparisons using the fdr_bh function implemented in MatlabR2022a. All results are reported for two-tailed statistical tests, unless reported otherwise. We removed outliers from all variables in our analyses based on values that exceeded 3 standard deviations from the mean.

### fMRI data

#### Acquisition

All neuroimaging data were acquired at the Caltech Brain Imaging Center using a Siemens Trio 3.0 Tesla scanner outfitted with a 32-channel phased-array head-coil. Functional image acquisition for the social perception task and altruism task occurred on different days in separate sessions (average delay of 303 days). For both fMRI tasks, we acquired gradient echo T2*-weighted echo-planar images (EPIs; 60 slices, voxel resolution $2.5 \times 2.5 \times 2.5$ mm$^3$, TR = 700 ms, TE = 30 ms, flip angle = 53°, FOV = 200 mm, interleaved acquisition order, multi-band acceleration factor = 6). We collected 903 volumes for the social perception task and 925 volumes for the altruism task. For all participants, we also acquired a high-resolution anatomical T1-weighted image using a MEMP-RAGE sequence (208 slices, $0.9 \times 0.9 \times 0.9$ mm$^3$, matrix size $256 \times 256$, TR = 2.55 s, TI = 1.15 s, TE = 1.6, 3.5, 5.3, 7.1 ms with RMS echo combination, RAGE flip angle = 8°). Distortion correction data for the fMRI EPI acquisitions employed a pair of phase-encoding polarity reversed T2w SE-EPI images with identical geometry and EPI echo train timing to the T2*w EPI images (TR 4.8 s, TE 50 ms, flip angle 90°). For information on preprocessing of functional and structural brain data, see Appendix 4.

#### General linear model of brain responses in the social perception task

This analysis aimed to identify brain regions recruited during different judgments in the social perception task. For each participant, we estimated a general linear model (GLM) using a canonical hemodynamic response function and a 128-s high-pass cut-off filter to eliminate low-frequency drifts in the data. The GLM estimated three regressors of interest corresponding to condition-wise brain responses (implemented in different blocks) during merit inferences (deserves help?), need inferences (needs help?), and factual control inferences that did not require social inferences (used both hands?; *Figure 1A*). The three condition-wise regressors were defined by the onset time of the first target image and the offset of the final image of each block (two blocks per condition). The GLM also included eight regressors of no interest: six motion regressors, the framewise displacement (estimated during preprocessing of neuroimaging data using fMRIPrep, https://fmriprep.org/; *Esteban et al., 2018a*), and a session constant. The GLMs were implemented in MATLAB (R2018b) using the SPM12 toolbox (http://www.fil.ion.ucl.ac.uk/spm).

For each participant, we created several contrasts of interest: First, we identified brain regions activated during merit perception (contrast [merit – control]) and need perception (contrast [need – control]). Second, we estimated brain regions that are significantly more (or less) activated for need compared to merit perception (contrasts [merit – need]; [need – merit]). The respective contrast images were then used in four group-level analyses using simple $t$-tests as implemented in SPM12. Whole-brain maps were thresholded at $p < 0.001$ (voxel level) and FWE corrected at a cluster level of $p < 0.05$. Finally, we tested for brain regions that were reliably activated during both need and merit perceptions, by estimating a formal conjunction of both inference-specific brain maps ([need

– control] ∩ [merit – control]) at the group level. Both brain maps used in this conjunction analysis were thresholded at the cluster-forming voxel level of p < 0.001, FWE cluster-level corrected at p < 0.05.

We also estimated a supplemental GLM for fMRI data from the altruism task. Due to COVID-19-related interruptions, only 25 participants from the sample that performed the social perception task also completed the fMRI altruism task. Given the limited sample size and noise level of fMRI data, we decided to focus solely on the behavior in the altruism task to address our research objectives.

## Multivariate decoding of the inference condition in the social perception task

We also performed multivariate decoding analysis on the brain data collected in the social perception task. We first estimated another GLM for each participant (using SPM12). The GLM was identical to the one used for the univariate analysis with two exceptions: first, the model was estimated on non-smoothed brain data. Second, instead of collapsing both task blocks per condition into one regressor of interest, the GLM modeled each block of the social perception task separately, yielding six regressors of interest ([condition: merit, need, control] × [block: 1, 2]).

The multivariate pattern analysis aimed to identify localized activation patterns that reliably decoded the perception condition (merit, need, control) in the social perception task. To ensure an unbiased analysis of the neural activation patterns throughout the whole brain, a 'searchlight' approach was used (*Kriegeskorte et al., 2006*; *Haynes et al., 2007*). Given that this approach does not depend on a priori assumptions about informative brain regions or prior voxel selection, the problem of circular analysis (or 'double dipping') can be avoided (*Kriegeskorte et al., 2009*). We implemented three separate decoding analyses: merit versus control, need versus control, merit versus need. All three analyses used a similar analysis approach, which we illustrate using the example of the decoding of need versus merit inferences below.

We used the same searchlight approach as in prior work (*Tusche et al., 2016*). For each participant, a sphere with a radius of 4 voxels was created around every voxel $v_i$ of the measured brain volume (*Tusche et al., 2014*; *Kahnt et al., 2011*; *Wisniewski et al., 2015*). For each sphere, we investigated whether the local pattern of activation predicted the inference condition (e.g., need vs. merit). For every task block, parameter estimates from the GLM (see above) were extracted for each of the $N$ voxels in the sphere around voxel $v_i$ and transformed into an $N$-dimensional pattern vector. In total, for the binary decoding analysis, we created four pattern vectors for each sphere (e.g., for a searchlight analysis decoding of need vs. merit, we extracted two block-wise pattern vectors for need, and two for merit). The first two of the pattern vectors (one per condition) were used for the training (training dataset) of a linear support vector machine classifier with a fixed regularization parameter C = 1 (implemented using libSVM operated in MATLAB, https://www.csie.ntu.edu.tw/~cjlin/libsvm/). This provided the basis for the subsequent classification of the remaining two pattern vectors (one per condition) that were not used for training. The procedure was then repeated by training on the last two pattern vectors and testing on the first two pattern vectors (yielding a twofold cross-validation). The amount of condition-related information of the spatial activation pattern of each spherical cluster was represented by the average decoding accuracy across both cross-validation steps and was assigned to the central voxel $v_i$ of the cluster.

The described classification was performed for all spherical clusters created around every measured voxel, resulting in a three-dimensional map of average classification accuracies for each participant. These participant-specific accuracy maps were then spatially smoothed with an isotropic Gaussian kernel of 6 mm full-width half-maximum. Finally, a standard second-level statistical analysis was performed to identify brain regions that allowed classifying the perceptual condition (e.g., whether individuals performed need or merit judgments) across participants ($n$ = 44). More specifically, individual accuracy maps were submitted to a one-sample $t$-test and contrasted against chance level (as implemented in SPM12). Since the classification was based on two alternatives (e.g., merit vs. need), chance level was 50%. Only regions passing a stringent statistical threshold (p < 0.001 at the voxel level, p < 0.05 FWE corrected at the cluster level) and showing significant decoding accuracies above chance were considered relevant for information encoding (*Haynes et al., 2007*; *Soon et al., 2008*). Results for the multivariate decoding analyses are summarized in *Supplementary file 2*.

## Acknowledgements

This research was supported by funding from NIMH Conte Center P50 MH094258 (CAH, AT), the Natural Sciences and Engineering Research Council of Canada RGPIN-2019-04329 (AT), and the Social Sciences and Humanities Research Council of Canada 435-2016-1274 (CAH). Computations were performed on the Niagara supercomputer at the SciNet HPC Consortium. SciNet is funded by Innovation, Science and Economic Development Canada; the Digital Research Alliance of Canada; the Ontario Research Fund: Research Excellence; and the University of Toronto. We sincerely thank Julian Michael Tyszka and Tim Armstrong for their support during data collection and Jill Jacobson for helpful comments on the manuscript.

## Additional information

### Funding

| Funder | Grant reference number | Author |
| --- | --- | --- |
| National Institute of Mental Health | Conte Center P50 MH094258 | Cendri A Hutcherson Anita Tusche |
| Natural Sciences and Engineering Research Council of Canada | RGPIN-2019-04329 | Anita Tusche |
| Social Sciences and Humanities Research Council of Canada | 435-2016-1274 | Cendri A Hutcherson |

The funders had no role in study design, data collection and interpretation, or the decision to submit the work for publication.

### Author contributions

Lisa M Bas, Formal analysis, Investigation, Visualization, Writing – original draft, Writing – review and editing; Ian D Roberts, Conceptualization, Software, Formal analysis, Investigation, Methodology, Writing – original draft, Project administration; Cendri A Hutcherson, Anita Tusche, Conceptualization, Supervision, Funding acquisition, Writing – original draft, Writing – review and editing

### Author ORCIDs

Lisa M Bas ![ORCID] https://orcid.org/0000-0003-0376-0059
Cendri A Hutcherson ![ORCID] https://orcid.org/0000-0002-4441-4809
Anita Tusche ![ORCID] https://orcid.org/0000-0003-4180-8447

### Ethics

All participants provided written informed consent and consent to publish, according to a protocol approved by the Institutional Review Board of the California Institute of Technology (#21-0692D).

Reviewer #1 (Public Review): https://doi.org/10.7554/eLife.92539.3.sa1
Reviewer #2 (Public Review): https://doi.org/10.7554/eLife.92539.3.sa2
Reviewer #3 (Public Review): https://doi.org/10.7554/eLife.92539.3.sa3
Author response https://doi.org/10.7554/eLife.92539.3.sa4

## Additional files

### Supplementary files

MDAR checklist

Supplementary file 1. Estimates of the computational model of social perception (hyper-mean parameter estimates, means of the posterior distributions with 95% Highest Density Interval, HDI).

Supplementary file 2. Whole-brain searchlight decoding of need, merit, and control inferences (social perception task).

Supplementary file 3. Brain regions activated during merit perceptions (social perception task) that reflect individual differences in merit sensitivity ($S_{merit}$) estimated in the computational model of social perception (in the n = 25 participants with overlapping altruistic choice task data).

Supplementary file 4. Logistic mixed models predicting generous choice (0/1).

Supplementary file 5. Model-estimated weights of choice-relevant attributes ($w_{self}$, $w_{other}$, $w_{fairness}$) and drift intercept bias ($w_0$) in the altruism task at the participant level (computational model of altruistic choice, n = 28).

Supplementary file 6. Hyper-mean parameter estimates (computational model of altruistic choice).

## Data availability

Functional imaging data from the social perception task and the altruism task are available at The Neurobiology of Social Decision-Making: Social Inference and Context collection. Raw behavioral data for both tasks, ROI masks, computational modeling data used for analysis and to create figures, and the newly created social perception task (including licensing details) are deposited on the Open Science Framework. Analysis scripts for the first-level GLMs and code for computational models are also available on OSF. FMRI data were preprocessed using the open-source fMRIPrep analysis pipeline (see hyperlinks in the manuscript). Source code underlying the supplemental MVPA analysis is openly available at *Hebart et al., 2014*.

The following dataset was generated:

| Author(s) | Year | Dataset title | Dataset URL | Database and Identifier |
|---|---|---|---|---|
| Roberts I, Hutcherson C, Tusche A, Adolphs R, Bas LM | 2019 | A neurocomputational account of the link between social perception and social action | https://osf.io/4u5vs/ | Open Science Framework, 4u5vs |

The following previously published datasets were used:

| Author(s) | Year | Dataset title | Dataset URL | Database and Identifier |
|---|---|---|---|---|
| Adolphs R | 2017 | Altruism Need Merit, The Neurobiology of Social Decision-Making: Social Inference and Context | https://nda.nih.gov/experiment.html?id=1320&collectionId=2643 | National Institute of Mental Health Data Archive, 2643 |
| Adolphs R | 2017 | Level of Inference Three (LOI3) | https://nda.nih.gov/experiment.html?id=1174&collectionId=2643 | National Institute of Mental Health Data Archive, 1174 |

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

# Appendix 1

## Economic exchange task to manipulate partner's merit in the altruism task

Before the altruism task, all participants completed a separate behavioral exchange game (sequential iterated Prisoner's Dilemma, modified from *Singer et al., 2004*; *Singer et al., 2006*). Participants were first informed about the game's overall structure, as follows: two anonymous players each start with 10 points. One player is randomly selected to act first and chooses how many points to send to the other player. Each sent point is tripled (e.g., if the player sends 10 points, the other player receives 30). The second player is then informed of the first player's choice and chooses how many of their total points to send in return (again, anything sent is tripled). Total joint earnings are maximized if each player sends the maximum number of points to the other player. Importantly, the second player can decide not to send any points back at the expense of the first player. Participants were informed that they would be paired with a different person (ostensibly another participant) for each trial. After the instruction, all participants were 'randomly' assigned to act as the second player. Participants completed 10 trials where each first player sent one of the following amounts: 1, 3, 5, 5, 7, 7, 9, 9, 10, and 10. The ordering of these trials was randomized across participants.

We implemented the following approach to manipulate the perceived merit of three anonymous players (who would act as partners in the subsequent altruism task). After playing the game themselves, participants received feedback about the choices of two other players and had to guess the choice of a third player. Specifically, participants were informed that these three individuals had played the sequential prisoners' dilemma at the same time as the participant but in different rooms. Participants were informed that player 1 behaved uncooperatively ('Sent 0 out of 10, which tripled to become 0'; indicating low merit), and player 2 behaved cooperatively ('Sent 10 out of 10, which tripled to become 30'; indicating high merit). As a control condition, no information about player 3 was provided (unknown merit); instead, participants were asked to guess that player's choices. Participants were then informed that they would play the next task (altruism task) in the scanner with those three partners, represented by three colored geometric shapes.

## Appendix 2

### Sanity check confirming enhanced perceptual sensitivity scores in task-relevant blocks of the social perception task

In the social perception task, we estimated three sensitivity parameters ($S_{merit}$, $S_{need}$, and $S_{control}$) independently for all three conditions (i.e., 9 sensitivity parameters total; 3 estimates × 3 conditions), irrespective of whether the judgment was task-relevant or not in the block. This allowed for the possibility that evidence relevant to a specific perceptual 'quantity' might be sampled in contexts where it was not explicitly required and might influence judgments even when it was irrelevant. That said, we would expect larger perceptual sensitivity estimates in task-relevant settings (e.g., enhanced merit sensitivity in merit blocks of the social perception task). To confirm this assumption (which in essence serves as a manipulation check), we evaluated the effects of experimental conditions on the estimated sensitivity values for the hyper-mean parameters in our computational model of social perception. Specifically, we compared parameter posteriors and converted the probabilities to two-tailed tests.

Perceptual sensitivity estimates differed between the three conditions, as would be expected. In each of the blocks, the task-relevant sensitivity estimates (e.g., $S_{need}$ in the need condition) were significantly greater than they were in either of the other conditions (e.g., $S_{need}$ in the merit and factual control blocks; all p's < 0.0001). Furthermore, $S_{control}$ did not differ between the need and merit conditions (p = 0.21), and $S_{merit}$ did not differ between the need and control conditions (p = 0.46). However, $S_{need}$ was significantly more positive in the merit block relative to the control conditions (p = 0.02).

Finally, participants exhibited a perceptual bias (overall drift *bias*) towards the 'yes' response in the merit condition (suggesting that, on average, people tend to perceive others as deserving) but not in the need or control conditions (see **_Supplementary file 1_**). The bias $w_0$ was significantly more negative in the need condition than in the merit condition (p = 0.002). The bias $w_0$ in the control condition was not significantly different from the need (p = 0.06) or merit conditions (p = 0.10) using two-tailed statistical tests.

## Appendix 3

## Model fitting procedure

Above, we introduced the computational behavioral models of social perception and altruistic choice. Here, we provide further details on the model fitting procedure. For both the social perception task and the altruism task, we identified the best-fitting parameter values by estimating their posterior distributions with hierarchical Bayesian models using differential evolution Markov chain Monte Carlo (DE-MCMC) sampling (*Holmes and Trueblood, 2018*; *Turner et al., 2013*). Specifically, we used an analytic solution (*Navarro and Fuss, 2009*) to calculate the likelihood of the observed data (i.e., choices and reaction times [RTs]) given a combination of parameter values and used this likelihood to construct a Bayesian estimate of the posterior distribution of the likelihood of the parameter values given the data. To maximize the amount of data used for fitting our model, we included trials in which participants did not respond before response deadlines by estimating the probability that simulations would fail to generate a choice within this time frame. Before model fitting, we removed a few trials where participants made a response in less than 200 ms (0% and 0.02% of trials in the social perception task and altruism task, respectively).

At the participant level of our hierarchical model for the *social perception task*, we fit seven parameters [*bias*, $S_{need}$, $S_{merit}$, $S_{control}$, *z*, *a*, *ndt*] for each of the task conditions (need/merit/control judgments) simultaneously. Specifically, the drift sensitivity parameters were estimated separately for each of the three task conditions (merit, need, control) and, thus, their values were independent across conditions. However, because we expected a participant's thresholds, starting biases, and non-decision times to be correlated across conditions, we used a method involving a 'baseline and difference' approach, in which one parameter specified the value obtained in the baseline/control (hands) condition, and two additional parameter governed the *difference* between the control value for that parameter and the specific task condition (i.e., merit − baseline, need − baseline). Thus, we obtained 21 parameter values per participant (7 free parameters × 3 conditions). At the population level of our model, we obtained estimates of the hyper means and hyper standard deviations for each of these parameters, resulting in 42 hyper parameters.

At the participant level of our hierarchical model for the *altruism task*, we fit three parameters [*z*, *a*, *ndt*] that were fixed across the need × merit conditions. In addition, we used a combination of four parameters [$w_0$, $w_{self}$, $w_{other}$, $w_{fairness}$] that were allowed to vary across conditions to describe the value-based evidence accumulation (drift) term in the model. For these parameters, we estimate only the independent main effects for need and merit, since another model that allowed for need × merit interactions did not improve model fit. Although the four weight parameters [$w_0$, $w_{self}$, $w_{other}$, $w_{fairness}$] were allowed to vary across the six combinations of need and merit, we also assumed that each of these parameters might be related within a single participant. Thus, we again took a 'baseline and difference' approach to capture how weight parameters differed across conditions. More specifically, our DDM model estimated four separate drift parameters for each attribute: (1) baseline sensitivity for the unknown partner; (2) an additive term related to difference in the need level ($\Delta_{Need}$, high = +1, low = −1); (3) an additive term indicating high merit partner trials, coded as 1/0; and (4) an additive term indicating low merit partner trials, coded as 1/0. For ease of comparison, when reporting the estimates for the main effect of need, we recode parameter estimates for high need as (baseline + $\Delta_{Need}$, see 1 and 2) and low need as (baseline − $\Delta_{Need}$, see 1 and 2). Thus, in total, we obtained 19 parameter values per participant. At the population level of our model we obtained estimates of the hyper means and hyper standard deviations for each of these parameters, resulting in 38 hyper parameters.

In total, we ran 3 × *k* chains in parallel for each model, where k is the number of participant-level parameters per participant (i.e., social perception task: 63 chains; altruism task: 57 chains) (*Holmes and Trueblood, 2018*). To preserve within-individual consistency for select parameter values (see above), we fit all conditions simultaneously. Additionally, we restricted possible parameter values as shown in *Appendix 3—table 1*. Given these constraints, we employed a transformation in parameter sampling to ensure that the prior distributions of the hyper-mean model parameters were truly uniform (noninformative) across the specified range in each condition.

Specifically, for the social perception task, the DE-MCMC sampler sampled three MCMC model parameters each for *ndt*, *a*, and *z*—one for each condition (i.e., need, merit, and control). To preserve within-participant consistency across the three conditions while ensuring that an inverse probit

transformation would result in a uniform distribution across values of [0,1], we computed the difference between one MCMC parameter and the sum of the other two (e.g., $ndt_{need} - (ndt_{merit} + ndt_{control})$) and then transformed this value via an inverse probit transformation. In this way, by specifying the priors of the MCMC hyper-mean model parameters as normal distributions with mean of 0 and variance of 1/3, $N(0, 1/3)$, their combination and subsequent inverse probit transformation yielded a uniform distribution across values of [0, 1] for parameters in each condition. These transformed values were then scaled by their respective functions, $f_{sc}(x)$, to the range of values as seen in *Appendix 3—table 1* to derive the model parameters. The priors of the MCMC hyper standard deviation parameters were specified as gamma distributions $\Gamma[1, 1]$.

For the altruism task, the DE-MCMC sampler sampled MCMC model drift weight ($w$) parameters composed of stability ($M_{params}$) and effect of need and merit manipulations (change across conditions; $\delta_{params}$). In the case of need-related effects, the sum of these MCMC model parameters $M_{params} + \delta_{params} \times$ Condition (effects coded as high need = 1; low need = –1) is then transformed via an inverse probit transformation. The priors of the MCMC hyper-mean model parameters for both $M_{params}$ and $\delta_{params}$ were specified as normal distributions with mean of 0 and variance of 0.5, $N(0, 0.5)$, because inverse probit transformations of the sum of two normal distributions with mean 0 and variance 0.5 yields a uniform distribution across values of [0, 1] for parameters in both conditions. These transformed values were then scaled by their respective functions, $f_{sc}(x)$, to the range of values as seen in *Appendix 3—table 1* to derive the diffusion model parameters. For the high and low merit conditions, the corresponding drift weight parameter was added following probit transformation. The priors of the MCMC hyper standard deviation parameters were specified as gamma distributions $\Gamma[1, 1]$.

To construct the estimated posterior distributions of each parameter, we sampled 5000 iterations after an initial burn-in period (burn-in of 15,000 for the social perception task, 10,000 for the altruism task). We then thinned these 5000 samples by only keeping every fifth sample, leaving 1000 samples per chain. For each iteration, the DE-MCMC algorithm proposes a new set of parameter values for each chain based on the scaled difference between two other randomly selected chains (*Turner et al., 2013*). The scaling for the difference between parameter values was $\gamma = 2.38/\sqrt{2d}$, where $d$ is the number of parameters in the parameter space (e.g., for the social perception task, 42 at the population level and 21 at the participant level) (*Turner et al., 2013*). At the hyper-level, parameter proposals were blocked such that one new parameter value was proposed at a time. At the participant level, parameter proposals were unblocked (*Turner et al., 2013*). The newly proposed parameter values are then evaluated by the Metropolis–Hastings algorithm for inclusion in the posterior distribution. Additionally, we implemented a probabilistic migration step, $\alpha = 0.01$, at each MCMC step instead of the differential evolution to improve chain mixing and convergence toward the high probability density region of the posterior distribution of parameters. The migration step cycles the positions of a subset of chains ($N_{migrate}$ uniformly sampled from the total number of chains) such that the positions of chains were compared against $i + 1, i + 2, \ldots j, i$ and evaluated based on the Metropolis–Hastings algorithm (*Turner et al., 2013*). All chains were assessed to have converged with the Gelman–Rubin statistic, $R$-hat < 1.1. An individual participant's parameters were estimated as the mean of the corresponding participant-level posterior distributions.

To assess the fits of extracted model parameters in predicting behavior, we used the fitted computational parameters for each participant to obtain model-predicted choice rates and median RTs. We assessed the model's ability to capture between-participant variability by correlating the observed and predicted choice rates and median RTs and then conducting a one-sample $t$-test on the choice and RT correlation coefficients.

**Appendix 3—table 1.** Scaling function, $f_{sc}(x)$, and resulting bounded range for each computational parameter in the model for the social perception task (left) and altruism task (right).

| Parameter | Social perception task | | Altruism task | |
|---|---|---|---|---|
| | $f_{sc}(x)$ | Range | $f_{sc}(x)$ | Range |
| All drift parameters ($S$, $w$) | $x \times 10 - 5$ | [−5, 5] | $x \times 10 - 5$ | [−5, 5] |
| $z$ | $x$ | [0, 1] | $x$ | [0, 1] |

*Appendix 3—table 1 Continued on next page*

*Appendix 3—table 1 Continued*

| | Social perception task | | Altruism task | |
|---|---|---|---|---|
| a | x × 9.999 − 0.001 | [0.001, 10.0] | x × 9.999 − 0.001 | [0.001, 10.0] |
| ndt | x × 2 | [0, 2] | x × 2 | [0, 2] |

Note. *S* = sensitivity, *w* = drift rates, *a* = difference between barriers, *z* = starting bias, *ndt* = non-decision time.

## Appendix 4

### Preprocessing

Preprocessing of all functional and structural brain data was performed using fMRIPrep 20.2.3 (RRID:SCR_016216) (*Esteban et al., 2018a*; *Esteban et al., 2018b*), which is based on Nipype 1.6.1 (*Gorgolewski et al., 2011*; *Gorgolewski, 2018*) (RRID:SCR_002502).

### Anatomical data preprocessing

A total of 1 T1-weighted (T1w) images were found within the input BIDS dataset. The T1-weighted (T1w) image was corrected for intensity nonuniformity (INU) with N4BiasFieldCorrection (*Tustison et al., 2010*), distributed with ANTs 2.3.3 (*Avants et al., 2008*) (RRID:SCR_004757), and used as T1w reference throughout the workflow. The T1w reference was then skull stripped with a Nipype implementation of the antsBrainExtraction.sh workflow (from ANTs), using OASIS30ANTs as target template. Brain tissue segmentation of cerebrospinal fluid (CSF), white-matter (WM), and gray-matter (GM) was performed on the brain-extracted T1w using fast (FSL 5.0.9, RRID:SCR_002823; *Zhang et al., 2001*). Brain surfaces were reconstructed using recon-all (FreeSurfer 6.0.1, RRID:SCR_001847, *Dale et al., 1999*), and the brain mask estimated previously was refined with a custom variation of the method to reconcile ANTs- and FreeSurfer-derived segmentations of the cortical gray-matter of Mindboggle (RRID:SCR_002438; *Klein et al., 2017*). Volume-based spatial normalization to two standard spaces (MNI152NLin6Asym, MNI152NLin2009cAsym) was performed through nonlinear registration with antsRegistration (ANTs 2.3.3), using brain-extracted versions of both T1w reference and the T1w template. The following templates were selected for spatial normalization: FSL's MNI ICBM 152 nonlinear 6th Generation Asymmetric Average Brain Stereotaxic Registration Model (*Evans et al., 2012*) [RRID:SCR_002823; TemplateFlow ID: MNI152NLin6Asym], ICBM 152 Nonlinear Asymmetrical template version 2009c [RRID:SCR_008796; TemplateFlow ID: MNI152NLin2009cAsym] (*Fonov et al., 2009*).

### Functional data preprocessing

For each of the BOLD runs (1 for the social perception task, 5 for the altruism task) found per participant (across all tasks and sessions), the following preprocessing was performed. First, a reference volume and its skull-stripped version were generated by aligning and averaging 1 single-band references (SBRefs). A B0-nonuniformity map (or fieldmap) was estimated based on two (or more) echo-planar imaging (EPI) references with opposing phase-encoding directions, with 3dQwarp (AFNI 20160207) (*Cox and Hyde, 1997*). Based on the estimated susceptibility distortion, a corrected EPI reference was calculated for a more accurate co-registration with the anatomical reference. The BOLD reference was then co-registered to the T1w reference using bbregister (FreeSurfer) which implements boundary-based registration (*Greve and Fischl, 2009*). Co-registration was configured with six degrees of freedom. Head-motion parameters with respect to the BOLD reference (transformation matrices, and six corresponding rotation and translation parameters) are estimated before any spatiotemporal filtering using mcflirt (FSL 5.0.9 *Jenkinson et al., 2002*). BOLD runs were slice-time corrected using 3dTshift from AFNI 20160207 (RRID:SCR_005927) (*Cox and Hyde, 1997*). First, a reference volume and its skull-stripped version were generated using a custom methodology of fMRIPrep. The BOLD time-series (including slice-timing correction when applied) were resampled onto their original, native space by applying a single, composite transform to correct for head-motion and susceptibility distortions. These resampled BOLD time-series will be referred to as preprocessed BOLD in original space, or just preprocessed BOLD. The BOLD time-series were resampled into several standard spaces, correspondingly generating the following spatially normalized, preprocessed BOLD runs: MNI152NLin6Asym, MNI152NLin2009cAsym. First, a reference volume and its skull-stripped version were generated using a custom methodology of fMRIPrep. Automatic removal of motion artifacts using independent component analysis (ICA-AROMA *Pruim et al., 2015*) was performed on the preprocessed BOLD on MNI space time-series after removal of non-steady state volumes and spatial smoothing with an isotropic, Gaussian kernel of 6 mm full-width half-maximum. Corresponding 'non-aggresively' denoised runs were produced after such smoothing. Additionally, the 'aggressive' noise regressors were collected and placed in the corresponding confounds file. Several confounding time-series were calculated based on the preprocessed BOLD: framewise displacement (FD), DVARS, and three region-wise global signals. FD was computed using two formulations following Power (*Power et al., 2014*) (absolute sum of relative motions) and Jenkinson (*Jenkinson et al., 2002*) (relative root mean square displacement between affines). FD and DVARS are calculated for each functional run, both using their implementations in Nipype (following the definitions by *Power et al.,*

*2014*). The three global signals are extracted within the CSF, the WM, and the whole-brain masks. Additionally, a set of physiological regressors were extracted to allow for component-based noise correction (CompCor, *Behzadi et al., 2007*). Principal components are estimated after high-pass filtering the preprocessed BOLD time-series (using a discrete cosine filter with 128 s cut-off) for the two CompCor variants: temporal (tCompCor) and anatomical (aCompCor). tCompCor components are then calculated from the top 2% variable voxels within the brain mask. For aCompCor, three probabilistic masks (CSF, WM, and combined CSF + WM) are generated in anatomical space. The implementation differs from that of *Behzadi et al., 2007* in that instead of eroding the masks by 2 pixels on BOLD space, the aCompCor masks are subtracted a mask of pixels that likely contain a volume fraction of GM. This mask is obtained by dilating a GM mask extracted from the FreeSurfer's aseg segmentation, and it ensures components are not extracted from voxels containing a minimal fraction of GM. Finally, these masks are resampled into BOLD space and binarized by thresholding at 0.99 (as in the original implementation). Components are also calculated separately within the WM and CSF masks. For each CompCor decomposition, the k components with the largest singular values are retained, such that the retained components' time-series are sufficient to explain 50 percent of variance across the nuisance mask (CSF, WM, combined, or temporal). The remaining components are dropped from consideration. The head-motion estimates calculated in the correction step were also placed within the corresponding confounds file. The confound time-series derived from head-motion estimates and global signals were expanded with the inclusion of temporal derivatives and quadratic terms for each (*Satterthwaite et al., 2013*). Frames that exceeded a threshold of 0.5 mm FD or 1.5 standardised DVARS were annotated as motion outliers. All resamplings can be performed with a single interpolation step by composing all the pertinent transformations (i.e., head-motion transform matrices, susceptibility distortion correction when available, and co-registrations to anatomical and output spaces). Gridded (volumetric) resamplings were performed using antsApplyTransforms (ANTs), configured with Lanczos interpolation to minimize the smoothing effects of other kernels (*Lanczos, 1964*). Non-gridded (surface) resamplings were performed using mri_vol2surf (FreeSurfer).

