## [Editor Report · eLife assessment]

These **important** findings stand out from other similar studies via some **convincing** demonstration of behavioural and neural relationships between two helping tasks – one focusing more on social perception, one more on its influence on social behaviour – that were performed more than 300 days apart. The claims however would be stronger with a larger sample size.

---

## [Referee Report · Reviewer #1 (Public Review)]

Summary:

The authors conducted two tasks at 300 days separation. First, a social perception task, where Ps responded whether a pictured person either deserved or needed help. Second, an altruism task, where Ps are offered monetary allocations for themselves and a partner. Ps decide whether to accept, or a default allocation of 20 dollars each. The partners differed in perceived merit, such that they were highly deserving, undeserving or unknown. This categorisation was decided on the basis of a prisoners dilemma game the partner played beforehand. "Need" was also manipulated, by altering the probability that the partner must have their hand in cold water at the end of the experiment and this partner can use the money to buy themselves out. These two tasks were conducted to assess the perception of need/merit in the first instance, and how this relates to social behaviour in the second. fMRI data were collected alongside behavioural.

The authors present many analyses of behaviour (including DDM results) and fMRI. E.g., they demonstrate that they could decode across the mentalising network whether someone was making a need or deserving judgement vs control judgements but couldn't decode need vs deserving. And that brain responses during merit inferences (merit - control) systematically covaried with participants' merit sensitivity scores in the rTPJ. They also found relationships between behaviour and rTPJ in the altruism task. And that merit sensitivity in the perception task predicted influence of merit on social behaviour in the altruism task.

Strengths:

This manuscript represents a sensible model to predict social perceptions and behaviours, and a tidy study design with interesting findings. The introduction introduced the field especially brilliantly for a general audience.

Weaknesses:

These are small samples. This is especially the case for the correlational questions. The limitation is acknowledged, but does mean that we cannot conclude much from absent relationships, where the likelihood of Type II error is high.

Decoding analyses. The authors decode need vs merit, and need+merit vs control, not the content of these inferences. The logic of these analyses implies that there is a distributed representation of merit that does not relate to its content but is an abstracted version that applies to all merit judgements. However, these analyses are not central to the authors' aims and conclusions, so this is just a minor point.

---

## [Referee Report · Reviewer #2 (Public Review)]

When people help others is an important psychological and neuroscientific question. It has received much attention from the psychological side, but comparatively less from neuroscience. The paper translates some ideas from a social Psychology domain to neuroscience using a neuroeconomically oriented computational approach. In particular, the paper is concerned with the idea that people help others based on perceptions of merit/deservingness, but also because they require/need help. To this end, the authors conduct two experiments with an overlapping participant pool:

(1) A social perception task in which people see images of people that have previously been rated on merit and need scales by other participants. In a blockwise fashion, people decide to whether the depicted person (a) deserves help, (b) needs help, and (c) whether the person uses both hands (== control condition)

(2) In an altruism task, people make costly helping decisions by deciding between giving a certain amount of money to themselves or another person. It is manipulated how much the other person needs and deserves the money.

The authors use sound and robust computational modelling approach for both tasks using evidence accumulation models. They analyse behavioural data for both tasks, showing that the behaviour is indeed influenced, as expected, by the deservingness and the need of the shown people. Neurally, the authors use a block-wise analysis approach to find differences in activity levels across conditions of the social perception task. The authors do find large activation clusters in areas related to theory of mind. Interestingly, they also find that activity in TPJ that relates to the deservingness condition correlates with people's deservingness ratings while they do the task, but also with computational parameters related to helping others in the second task, the one that was conducted many months later. Also some behavioural parameters correlate across the two tasks, suggesting that how deserving of help others are perceived reflects a relatively stable feature that translates into concrete helping decisions later-on.

The conclusions of the paper are overall well supported by the data.

(1) I found that the modelling was done very thoroughly for both tasks. Overall, I had the impression that the methods are very solid with many supplementary analyses. The computational modelling is done very well.

(2) A slight caveat, however, regarding this aspect, is that, in my view, the tasks are relatively simplistic, so that even the complex computational models do not as much as they can in the case of more complex paradigms. For example, the bias term in the model seems to correspond to the mean response rate in a very direct way (please correct me if I am wrong).

(3) Related to the simple tasks: The fMRI data is analysed in a simple block-fashion. This is in my view not appropriate to discern the more subtle neural substrates of merit/need-based decision making or person perception. Correspondingly, the neural activation patterns (merit > control, need > control) are relatively broad and unspecific. They do not seem to differ in the classic theory of mind regions, that are the focus of the analyses.

(4) However, the relationship between neural signal and behavioural merit sensitivity in TPJ is noteworthy.

(5) The latter is even more the case, as the neural signal and aspects of the behaviour are correlated across subjects with the second task that is conducted much later. Such a correlation is very impressive and suggests that the tasks are sensitive for important individual differences in helping perception/behaviour.

(6) That being said, the number of participants in the latter analyses are at the lower end of the number of participants that are these days used for across-participant correlations.

---

## [Referee Report · Reviewer #3 (Public Review)]

Summary:

The paper aims at providing a neurocomputational account on how social perception translates in prosocial behaviors. Participants first completed a novel social perception task during fMRI scanning, in which were asked to judge the merit or need of people depicted in different situations. Second , a separate altruistic choice task was used to examine how the perception of merit and need influences the weights people place on themselves, others and fairness when deciding to provide help. Finally, a link between perception and action was drawn in those participants who completed both tasks.

Strengths:

The paper is overall very well written and presented, leaving the reader at ease when describing complex methods and results. The approach used by the author is very compelling, as it combines computational modeling of behavior and neuroimaging data analyses. Despite not being able to comment on the computational model, I find the approach used (to disentangle sensitivity and biases, for merit and need) very well described and derived from previous theoretical work. Results are also clearly described and interpreted.

Weaknesses:

In the social perception task, merit and need are evaluated by means of very different cues that rely on different cognitive processes (more abstract thinking for merit than need). Despite this limitation of the task, the authors were able to argue convincingly in the revised version about the solidity of their findings. Sample size is quite small for study 2, nevertheless the results provide convincing evidence.

---

## [Author Response]

The following is the authors’ response to the original reviews.

**Public Reviews:**

**Reviewer #1 (Public Review):**
Summary:The authors conducted two tasks at 300 days of separation. First, a social perception task, where Ps responded whether a pictured person either deserved or needed help. Second, an altruism task, where Ps are offered monetary allocations for themselves and a partner. Ps decide whether to accept, or a default allocation of 20 dollars each. The partners differed in perceived merit, such that they were highly deserving, undeserving, or unknown. This categorisation was decided on the basis of a prisoner's dilemma game the partner played beforehand. "Need" was also manipulated, by altering the probability that the partner must have their hand in cold water at the end of the experiment and this partner can use the money to buy themselves out. These two tasks were conducted to assess the perception of need/merit in the first instance, and how this relates to social behaviour in the second. fMRI data were collected alongside behavioural.The authors present many analyses of behaviour (including DDM results) and fMRI. E.g., they demonstrate that they could decode across the mentalising network whether someone was making a need or deserving judgement vs control judgement but couldn't decode need vs deserving. And that brain responses during merit inferences (merit - control) systematically covaried with participants' merit sensitivity scores in the rTPJ. They also found relationships between behaviour and rTPJ in the altruism task. And that merit sensitivity in the perception task predicted the influence of merit on social behaviour in the altruism task.Strengths:This manuscript represents a sensible model to predict social perceptions and behaviours, and a tidy study design with interesting findings. The introduction introduced the field especially brilliantly for a general audience.

Response: We are pleased that the reviewer found the model sensible and the findings interesting!Below, we respond to each of the reviewer’s comments/critiques.

Weaknesses:(1) The authors do acknowledge right at the end that these are small samples. This is especially the case for the correlational questions. While the limitation is acknowledged at the end, it is not truly acknowledged in the way that the data are interpreted. I.e. much is concluded from absent relationships, where the likelihood of Type II error is high in this scenario. I suggest that throughout the manuscript, authors play down their conclusions about absence of effects.

Response: We agree with the reviewer that the limitation of small samples should be adequately reflected in the interpretation of the data. We have therefore added cautionary language to the interpretation of the correlational effects in several places of the revised manuscript. For example, we now state: “However, this absence of effects for need ought to be interpreted with caution, given the comparatively small sample size.” (pg. 33) and “As mentioned above, we cannot rule out the possibility that null findings may be due to the comparatively small sample size and should be interpreted cautiously (also see discussion)” (pg. 34-35).

(2) I found the results section quite a marathon, and due to its length I started to lose the thread concerning the overarching aims - which had been established so neatly in the introduction. I am unsure whether all of these analyses were necessary for addressing the key questions or whether some were more exploratory. E.g. it's unclear to me what one would have predicted upfront about the decoding analyses.

Response: We acknowledge and share the reviewer’s concern about the length of the results section and potential loss of clarity. Regarding the decoding analyses, we want to clarify that they were conducted as a sanity check to compare against the results of the univariate analysis. We didn’t have apriori hypotheses regarding these supplemental decoding analysis. We have clarified this issue in the revised version of the manuscript and moved the decoding analyses fully to the supplemental material to streamline the main text. The remaining results reported in the manuscript are indeed all based on apriori, key questions (unless specified otherwise, for example, supplemental analyses for other regions of interest for the sake of completeness). The only exception is the final set of results (Neural markers of merit sensitivity predict merit-related behavioral changes during altruistic choice) which represent posthoc tests to clarify the role of activation in the right temporoparietal junction (rTPJ) in merit-related changes in other-regard in altruistic decisions. While we acknowledge that this is a complex paper, after careful consideration we couldn’t identify any other parts of the results section to remove or report in the supplemental material.

(3) More specifically, the decoding analyses were intriguing to me. If I understand the authors, they are decoding need vs merit, and need+merit vs control, not the content of these inferences. Do they consider that there is a distributed representation of merit that does not relate to its content but is an abstracted version that applies to all merit judgements? I certainly would not have predicted this and think the analyses raise many questions.

Response: We thank the reviewer for sharing their thoughts on the decoding analyses and agree that this set of analyses are intriguing, yet raise additional questions, such as the neural computations required to assess content. However, we wish to clarify that the way we view our current results is very much analogous to results obtained from studies of perception in other fields. For example, in the face perception literature, it is often observed that the fusiform face area is uniformly more active, not only when a face (as opposed to an object) is on the screen, but when a compound stimulus consistent of features of a face and other features (e.g. of objects) is on the screen, but participants are instructed to attend to and identify solely the face. Moreover, multivariate activity in the FFA (but not univariate activity) is sufficient to decode the identity of the face. We view the results we report in the manuscript as more akin to the former types of analyses, where any region that is involved in the computation is uniformly more active when attention is directed to judgment-specific features. Unfortunately, the present data are not sufficient to properly answer the latter questions, about which areas enable decoding of specific intensity or identity of merit-related content. Follow-up experiments with a more optimized design are needed. Although interesting, we thus refrain from further discussing the decoding analyses in the manuscript to avoid distracting from the main findings based on the univariate comparison of brain responses observed while participants make merit or need inferences in the social perception task.

**Reviewer #2 (Public Review):**
When people help others is an important psychological and neuroscientific question. It has received much attention from the psychological side, but comparatively less from neuroscience. The paper translates some ideas from a social Psychology domain to neuroscience using a neuroeconomically oriented computational approach. In particular, the paper is concerned with the idea that people help others based on perceptions of merit/deservingness, but also because they require/need help. To this end, the authors conduct two experiments with an overlapping participant pool:(1) A social perception task in which people see images of people that have previously been rated on merit and need scales by other participants. In a blockwise fashion, people decide whether the depicted person (a) deserves help, (b) needs help, and (c) whether the person uses both hands (== control condition).(2) In an altruism task, people make costly helping decisions by deciding between giving a certain amount of money to themselves or another person. How much the other person needs and deserves the money is manipulated.The authors use a sound and robust computational modelling approach for both tasks using evidence accumulation models. They analyse behavioural data for both tasks, showing that the behaviour is indeed influenced, as expected, by the deservingness and the need of the shown people. Neurally, the authors use a block-wise analysis approach to find differences in activity levels across conditions of the social perception task (there is no fMRI data for the other task). The authors do find large activation clusters in areas related to the theory of mind. Interestingly, they also find that activity in TPJ that relates to the deservingness condition correlates with people's deservingness ratings while they do the task, but also with computational parameters related to helping others in the second task, the one that was conducted many months later. Also, some behavioural parameters correlate across the two tasks, suggesting that how deserving of help others are perceived reflects a relatively stable feature that translates into concrete helping decisions later-on.The conclusions of the paper are overall well supported by the data.

Response: We thank the reviewer for the positive evaluation of our study and the comprehensive summary of our main findings. We would like to clarify, though, that we did originally collect fMRI data for the independent altruism task. Unfortunately, due to COVID-19-related interruptions, only 25 participants from the sample that performed the social perception task also completed the fMRI altruism task (see pg. 18). Given the limited sample size and noise level of fMRI data, we moved anything related to the neuroimaging data of the altruism task to the supplemental material (see Note S7) and decided to focus solely on the behavior of the altruism task to address our research objectives. We apologize for any confusion.

(1) I found that the modelling was done very thoroughly for both tasks. Overall, I had the impression that the methods are very solid with many supplementary analyses. The computational modelling is done very well.

Response: We are pleased that the reviewer found the computational model sensible.

(2) A slight caveat, however, regarding this aspect, is that, in my view, the tasks are relatively simplistic, so even the complex computational models do not do as much as they can in the case of more complex paradigms. For example, the bias term in the model seems to correspond to the mean response rate in a very direct way (please correct me if I am wrong).

Response. We agree that the Bias term relates to mean responding (although it is not the sole possibility: thresholds and starting default biases can also produce changes in mean levels of responding that, without the computational model, are not possible to dissociate). However, we think that the primary value of this parameter comes not from the analysis of the social judgment task (where the reviewer is correct that the bias relates in a quite straightforward way to the mean response rate), but in the relationship of this parameter to the un-contextual generosity response in the altruism task. Here, we find that this general bias term relates not to overall generosity, but rather to the overall weight given to others’ outcomes, a finding that makes sense if the tendency to perceive others as deserving overall yields an increase in overall attention/valuation of their outcomes. Thus, a simple finding in one task relates to a more nuanced finding in another. However, we agree it is important to acknowledge the point raised by the reviewer, and now do so on pg. 20: “It is worth noting that the Bias parameters are strongly associated with (though not the sole determinant of) the mean response rate.”

(3) Related to the simple tasks: The fMRI data is analysed in a simple block-fashion. This is in my view not appropriate to discern the more subtle neural substrates of merit/need-based decision-making or person perception. Correspondingly, the neural activation patterns (merit > control, need > control) are relatively broad and unspecific. They do not seem to differ in the classic theory of mind regions, which are the focus of the analyses.

Response: The social perception task is modified from a well-established social inference task (Spunt & Adolphs, 2014; 2015) designed to reliably localize the mentalizing network in the brain. As such, we acknowledge that it is not optimally designed to discern the intrinsic complexities of social perception, or the specific appraisals or computations that yield more or less perception (of need or merit) in a given context. Instead, it was designed to highlight regions that are more generally recruited for performing these social perceptions/inferences.

We heartily agree with the reviewer that it would be interesting and informative to analyze this task in a trial-wise way, with parametric variation in evidence for each image predicting parametric variation in brain activity. Unfortunately, the timing of this task is not optimal for this kind of an analysis, since trials were presented in rapid and blocked fashion. We were also limited in the amount of time we could devote to this task, since it was collected in conjunction with a number of other tasks as part of a larger effort to detail the neural correlates of social inference (reported elsewhere). Thus, we were not able to introduce the kind of jittered spacing between trials that would have enabled such analysis, despite our own wish to do so. We hope that this work will thus be a motivator for future work designed more specifically to address this interesting question, and now include a statement to this effect on pgs. 2223: “Future research may reveal additional distinctions between merit and need appraisals in trial-wise (compared to our block-wise) fMRI designs.”

References:

Spunt, R. P. & Adolphs, R. Validating the Why/How contrast for functional MRI studies of Theory of Mind. Neuroimage 99, 301-311, doi:10.1016/j.neuroimage.2014.05.023 (2014).

Spunt, R. P. & Adolphs, R. Folk explanations of behavior: a specialized use of a domain-general mechanism. Psychological Science 26, 724-736, doi:10.1177/0956797615569002 (2015).

(4) However, the relationship between neural signal and behavioural merit sensitivity in TPJ is noteworthy.

Response: We agree with this assessment and thank the reviewer for their positive assessment; we feel that linking individual differences in merit sensitivity with variance in TPJ activity during merit judgments is one of the key findings of the study.

(5) The latter is even more the case, as the neural signal and aspects of the behaviour are correlated across subjects with the second task that is conducted much later. Such a correlation is very impressive and suggests that the tasks are sensitive for important individual differences in helping perception/behaviour.

Response: Again, we share the reviewer’s impression that this finding is more noteworthy for appearing in tasks separated both by considerable conceptual/paradigmatic differences, and by such a long temporal distance. These findings make us particularly excited to follow up on these results in future research.

(6) That being said, the number of participants in the latter analyses are at the lower end of the number of participants that are these days used for across-participant correlations.

Response: We fully agree with this assessment. Unfortunately, COVID-related disruptions in data collection, as well as the expiration of grant funds due to the delay, severely limited our ability to complete assessments in a larger sample. Future research needs to replicate these results in a larger sample. We comment on this issue in the discussion on pg. 40. If the editor or reviewer has suggestions for other ways in which we could more fully acknowledge this, we would be happy to include them.

**Reviewer #3 (Public Review):**
Summary:The paper aims to provide a neurocomputational account of how social perception translates into prosocial behaviors. Participants first completed a novel social perception task during fMRI scanning, in which they were asked to judge the merit or need of people depicted in different situations. Secondly, a separate altruistic choice task was used to examine how the perception of merit and need influences the weights people place on themselves, others, and fairness when deciding to provide help. Finally, a link between perception and action was drawn in those participants who completed both tasks.Strengths:The paper is overall very well written and presented, leaving the reader at ease when describing complex methods and results. The approach used by the author is very compelling, as it combines computational modeling of behavior and neuroimaging data analyses. Despite not being able to comment on the computational model, I find the approach used (to disentangle sensitivity and biases, for merit and need) very well described and derived from previous theoretical work. Results are also clearly described and interpreted.

Response: We thank the reviewer for their positive comments regarding presentation, approach, and content.

Weaknesses:My main concern relates to the selection of the social perception task, which to me is the weakest point. Such weakness has been also addressed by the same authors in the limitation section, and related to the fact that merit and need are evaluated by means of very different cues that rely on different cognitive processes (more abstract thinking for merit than need). I wonder whether and how such difference can bias the overall computational model and interpretation of the results (e.g. ideal you vary merit and need to leave all other aspects invariant).

Response: We agree with the reviewer on the importance of future research to more fully unpack the differences in this task, and develop better ways to manipulate need and merit in more comparable fashion. However, we point out that the issue of differences in abstractness of cues for need and merit does not actually seem to have a strong influence on the parameters retrieved by the computational model. Participants seem to be equally sensitive to BOTH merit and need information, despite that information deriving from different sources, as evidenced by the fact that the magnitude of the sensitivity parameters for need and merit in the social judgment task were nearly identical, and not statistically distinguishable. Nor were other parameters related to non-decision time or threshold statistically different (see Supplemental Table S2). If our results were driven purely by differences in the difficulty or abstractness of these judgments, we would have expected to see some evidence of this in the computational model, in the form of longer non-decision times, higher thresholds, or both. We do not. Likewise, the neural underpinnings evoked by both need and merit perceptions in this task (in the mentalizing brain network) were comparable. This is not to say that there aren’t real differences in the cues that might signal these quantities in our social perception task - just that there is little direct evidence for this difference in computational parameters or evoked brain responses, and thus it is unlikely that our results (which rely on an analysis of computational parameters) are driven solely by computational model biases, or the inability of the model to adequately assess participant sensitivity to need as opposed to merit.

A second weakness is related to the sample size which is quite small for study 2. I wonder, given that study 2 fRMI data are not analyzed, whether is possible to recover some of the participants' behavioral results, at least the ones excluded because of bad MR image quality.

Response: We fully agree with the reviewer that increasing the sample size for the cross-task correlations would be desirable. Unfortunately, the current sample size already presents the maximum of ‘usable’ data; the approach suggested by the reviewer won’t affect the sample size. We used all participants whose behavioral data in the altruism task suggested they were performing the task in good faith and conscientiously.

Finally, on a theoretical note, I would elaborate more on the distinction of merit and need. These concepts tap into very specific aspects of morality, which I suspect have been widely explored. At the moment I am missing a more elaborate account of this.

Response: Need and merit are predominantly studied in separate lines of research (Molouki & Bartels),

1. so there is relatively little theoretical research on the distinction between the two. Consequently, Siemoneit (2023) states that the relation between the concepts of need and merit in allocative distributions remains diffuse. To emphasize the distinct concepts of morality in the introduction we have now added to pg. 3: “Need and deservingness (merit) are two distinct principles of morality. The need principle involves distributing resources to those who require them, irrespective of whether they have earned them, while the "merit principle" focuses on allocating resources based on individuals' deservingness, regardless of their actual need (Wilson, 2003).”

One of the added values of our paper to the research literature is in adding to the clarification of computational and neural underpinnings of broad concepts like merit and need. To highlight the latter point, we have added the following statement on pg. 5 to the manuscript: “Examining need and merit concurrently in this task will also help clarify the computational and neural underpinnings of related, but distinct concepts, distinguishing between them more effectively.”

References:

Molouki, S., & Bartels, D. M. (2020). Are future selves treated like others? Comparing determinants and levels of intrapersonal and interpersonal allocations. Cognition, 196, 104150.

Siemoneit, A. (2023). Merit first, need and equality second: hierarchies of justice. International Review of Economics, 70(4), 537-567.

Wilson, C. (2003). The role of a merit principle in distributive justice. The Journal of ethics, 7, 277-314.

**Recommendations for the authors:**

**Reviewer #1 (Recommendations For The Authors):**
I acknowledge the difficulty with respect to recruitment, especially in the age of covid, but is it possible for the authors to collect larger samples for their behavioural questions via online testing? Admittedly, I'm sure they don't want to wait 300 days to have the complete dataset, but I would be in favour of collecting a sample in the hundreds on these behavioural tasks, completed at a much shorter separation (if any). I believe this would strengthen the authors' conclusions considerably if they could both replicate the effects they have and check these null effects in a sample where they could draw conclusions from them. Indeed, Bayesian stats to provide evidence for the null would also help here.

Response: We share the reviewer’s desire to see these results replicated (ideally in a sample of hundreds of participants). We have seriously considered the possibility of trying to replicate our results online, even before submitting the first version of the paper. However, it is difficult to fully replicate this paradigm online, given the elaborate story and context we engaged in to convince participants that they were playing with real others, as well as the usage of physical pain (Cold Pressor Task) for the need manipulation in the altruism task. Moreover, given comments by this reviewer that the results are already a little long, adding a new, behavioral replication would likely only add to the memory burden for the reader. We have thus opted not to include a replication study in the current work. However, we are actively working on a replication that can be completed online, using a modified experimental paradigm and different ways to manipulate need and merit. Because of the differences between that paradigm and the one described here, which would require considerable additional exposition, we have opted not to include the results of this work in the current paper. We hope to be able to publish this work as a separate, replication attempt in the future.

Given the difficulty of wading through the results section while keeping track of the key question being answered, I would suggest moving any analyses that are less central to the supplementary. And perhaps adding some more guiding sentences at the start and end of each section to remind the reader how each informs the core question.

Response: We deliberated for quite some time about what results could be removed, but in the end, felt that nearly all results that we already described need to be included in the paper, since each piece of the puzzle contributes to the central finding (relating parameters and behavior to neural and choice data across two separate tasks). However, we did move the decoding analysis results to the supplemental (see point below). We also take the reviewers point that the results can be made clearer. We thus have worked to include some guiding sentences at the start and end of sections to remind the readers how each analysis informs the core questions.

I think it needs unpacking more for the reader what they should conclude from the significant need+merit vs control decoding analyses, and what they would have expected in terms of cortical representation from the decoding analyses in general.

Response: We agree with the reviewer that given the decoding results position in the main manuscript it would need unpacking. After considering the reviewer's prior suggestion, we have reevaluated the placement of these supplemental results. Consequently, we have relocated it to the supplemental materials, as it was deemed less relevant to directly addressing the core research questions in the main manuscript. On pg. 23, the main manuscript now only states “We also employed supplemental multivariate decoding analyses (searchlight analysis 85-87), as commonly used in social perception and neuroscience research 7,58,82,88,89, corroborating our univariate findings (see Supplemental Note S6, Supplemental Table S10).”

**Reviewer #2 (Recommendations For The Authors):**
(1) I would suggest moving information on how the computational models were fitted to the main text.

Response: The computational models are a key element of the paper and we deliberated about the more central exposure of the description of how the models were fitted in the main manuscript. However, we are concerned about the complexity and length of the article, which requires quite a lot from readers to keep in mind (as also commented on by reviewer 1). Those readers who are particularly interested in details of model fitting can still find an extensive discussion of the procedures we followed in the supplements. We thus have opted to retain the streamlined presentation in the main manuscript. However, if the editor feels that including the full and extensive description of model fitting in the main paper would significantly improve the flow and exposition of ideas, we are happy to do so.

(2) For the fMRI analyses: Could it be worth analysing the choices in the different conditions? They could be modelled as a binary regressor (yes/no) and this one might be different across conditions (merit/need/hands). Maybe this won't work because of the tight trial timeline, but it could be another avenue to discern differences across fMRI conditions.

Response: We thank the reviewer for this interesting suggestion! Unfortunately, the block design and rapid presentation of stimuli within each condition make it challenging to distinguish the different choices (within or across conditions). While we see the merit in the suggested analytical approach (in fact, we discussed it before the initial submission of the article), it would require some modifications of the task structure (e.g., longer inter-trial-intervals between individual stimuli) and an independent replication fMRI study. We were not able to have such a long inter-trial interval in the original design due to practical constraints on the inclusion of this paradigm in a larger effort to examine a wide variety of social judgment and inference tasks. We hope to investigate this kind of question in greater detail in future fMRI work.

(3) The merit effects seem to be more stable across time than the need conditions. Would it be worthwhile to test if the tasks entailed a similar amount of merit and need variation? Maybe one variable varied more than the other in the task design, and that is why one type of effect might be stronger than the other?

Response: We thank the reviewer for drawing attention to this important point. We used extensive pilot testing to select the stimuli for the social perception task, ensuring an overall similar amount of need and merit variation. For example, the social perception ratings of the independent, normative sample suggest that the social perception task entails a similar amount of need and merit variation (normative participant-specific percentage of yes responses for merit (mean ± standard deviation: 53.95 ± 13.87) and need (45.65 ± 11.07)). The results of a supplemental paired t-test (p = 0.122) indicate comparable SD for need and merit judgments. Moreover, regarding the actual fMRI participant sample, Figure S3 illustrates comparable levels of variations in need and merit perceptions (participant-specific percentage of yes responses for merit (56.70 ± 11.91) and need (48.69 ± 10.81) in the social perception task). Matching the results for the normative sample, the results of a paired t-test (p = 0.705) suggest no significant difference in variation between need and merit judgments. With respect to the altruism task, we manipulated the levels of merit and need externally (high vs. low).

**Reviewer #3 (Recommendations For The Authors):**
(1) It would be good to provide the demographics of each remaining sample.

Response: We appreciate the attention to detail and agree with the reviewer’s suggestion. We have now added the demographics for each remaining sample to the revised manuscript.

(2) The time range from study 1 to study 2, is quite diverse. Did you use it as a regressor of no interest?

Response: We thank the reviewer for this interesting suggestion. We have examined this in detail in the context of our cross-task analyses (i.e., via regressions and partial correlations). Interestingly, variance in the temporal delay between both tasks does not account for any meaningful variation, and results don’t qualitatively change controlling for this factor.

For example, when we controlled for the delay between both separate tasks (partial correlation analysis), we confirmed that variance in merit sensitivity (social perception task) still reflected meritinduced changes in overall generosity (altruism task; p = 0.020). Moreover, we confirmed that variance in merit sensitivity reflected individuals’ other-regard (p = 0.035) and self-regard (p = 0.040), but not fairness considerations (p = 0.764) guiding altruistic choices. Regarding people’s general tendency to perceive others as deserving, we found that the link between merit bias (social perception task) and overall other-regard (p = 0.008) and fairness consideration (p = 0.014) (altruism task) holds when controlling for the time range (no significant relationship between merit bias and self-regard, p = 0.191, matching results of the main paper).

We refer to these supplemental analyses in the revised manuscript on ps. 33 and 35: “Results were qualitatively similar when statistically controlling for the delay between both tasks (partial correlations).”

(3) Why in study 1 a dichotomous answer has been used? Would not have been better (also for modeling) a continuous variable (VAS)?

Response: We appreciate the reviewer's thoughtful feedback. In Study 1, opting for a dichotomous response format in the social perception task (Figure 1a) was a deliberate methodological choice. This decision, driven by the study's model requirements, aligns with the common use of a computational model employing two-alternative forced choices ("yes" and "no") as decision boundaries. While drift– diffusion models for multiple-alternative forced-choice designs exist, our study's novel research questions were effectively addressed without their complexity. Finally, our model cannot accept continuous response variables as input unless they are transformed into categorical variables.

(4) In the fMRI analyses, when you assess changes in brain activity as a function of merit, I would control for need (and the other way round), to see whether such association is specific.

Response: Regarding the reviewer’s suggestion on controlling for need when assessing changes in brain activity as a function of merit, and vice versa, we would like to clarify the nature of our fMRI analyses in the social perception task. Our focus is on block-wise assessments (need vs. control, merit vs. control, need vs. merit blocks, following the fMRI task design from which our social perception task was modified from). We don’t assess changes in brain activity as a function of the level of perceived merit or need (i.e., “yes” vs. “no” trials within or across task blocks). Blocks are clearly defined by the task instruction given to participants prior to each block (i.e., need, merit, or control judgments). Thus, unfortunately, given the short inter-stimulus-intervals of each block, the task design is not optimal to implement the suggested approach.